# CD36 facilitates fatty acid uptake by dynamic palmitoylation-regulated endocytosis

Jian-Wei Hao[1,6], Juan Wang[1,2,6], Huiling Guo[1,6], Yin-Yue Zhao[1], Hui-Hui Sun[1], Yi-Fan Li[1], Xiao-Ying Lai[1], Ning Zhao[1], Xu Wang[2], Changchuan Xie[1], Lixin Hong[1], Xi Huang[1], Hong-Rui Wang [1,3], Cheng-Bin Li[4], Bin Liang[4], Shuai Chen[5✉] & Tong-Jin Zhao [1,2,3✉]

Fatty acids (FAs) are essential nutrients, but how they are transported into cells remains unclear. Here, we show that FAs trigger caveolae-dependent CD36 internalization, which in turn delivers FAs into adipocytes. During the process, binding of FAs to CD36 activates its downstream kinase LYN, which phosphorylates DHHC5, the palmitoyl acyltransferase of CD36, at Tyr91 and inactivates it. CD36 then gets depalmitoylated by APT1 and recruits another tyrosine kinase SYK to phosphorylate JNK and VAVs to initiate endocytic uptake of FAs. Blocking CD36 internalization by inhibiting APT1, LYN or SYK abolishes CD36-dependent FA uptake. Restricting CD36 at either palmitoylated or depalmitoylated state eliminates its FA uptake activity, indicating an essential role of dynamic palmitoylation of CD36. Furthermore, blocking endocytosis by targeting LYN or SYK inhibits CD36-dependent lipid droplet growth in adipocytes and high-fat-diet induced weight gain in mice. Our study has uncovered a dynamic palmitoylation-regulated endocytic pathway to take up FAs.

[1] State Key Laboratory of Cellular Stress Biology, School of Life Sciences, Xiamen University, Xiamen 361102 Fujian, China. [2] Institute of Metabolism and Integrative Biology, Fudan University, Shanghai 200438, China. [3] State-Province Joint Engineering Laboratory of Targeted Drugs from Natural Products, Xiamen University, Xiamen 361102 Fujian, China. [4] Center for Life Sciences, School of Life Sciences, Yunnan University, Kunming 650091, China. [5] State Key Laboratory of Pharmaceutical Biotechnology, Model Animal Research Center, Nanjing Biomedical Research Institute, Nanjing University, Pukou District, Nanjing 210061, China. [6] These authors contributed equally: Jian-Wei Hao, Juan Wang, Huiling Guo. ✉email: chenshuai@nju.edu.cn; zhaotj@fudan.edu.cn

Fatty acids (FAs) exert multiple functional roles, including serving as an energy source and as precursors for membrane synthesis and energy storage[1]. FAs and their derivatives can also function as signaling molecules to regulate multiple cellular activities. Despite the important roles of FAs, the first regulatory step for their utilization, i.e. transporting FAs across the plasma membrane, is not well understood.

Unlike glucose and amino acids, FAs are hydrophobic and have extremely low aqueous solubility at physiological pH 7.4, and they are usually bound with serum albumins and cytoplasmic FA binding proteins[2]. The hydrophobicity makes it difficult to track the movement of FAs. It was proposed that passive diffusion might be the predominant way to absorb FAs[3], but accumulating evidences support that protein-facilitated FA uptake is the key pathway in metabolic tissues including liver, adipose tissue, and muscle[4,5].

Among the proteins involved in FA uptake, CD36 is a major player in metabolic tissues[6,7]. It is primarily localized in and requires caveolae for its FA uptake activity[8–10]. CD36 accounts for 50% of the FA uptake in adipose tissues and muscle in mice[11]. Humans with CD36 deficiency also show significantly decreased FA uptake in heart[12], muscle, and adipose tissues[13]. CD36 has an FA binding pocket[14], and it was proposed to be a transporter[6]; however, a study using HEK293 cells shows that CD36 increases intracellular esterification but not translocation of FAs[15]. It still remains unknown about how exactly CD36 transports FAs across the plasma membrane.

Protein palmitoylation is catalyzed by a group of Asp-His-His-Cys (DHHC)-motif containing palmitoyl acyltransferases (DHHCs) by attaching a palmitoyl group to a thiol group of a cysteine residue[16,17]. Dynamic palmitoylation provides a key regulatory mechanism for subcellular localization and intracellular trafficking of proteins[18,19]. We have previously shown that palmitoylation of CD36 by DHHC4 and DHHC5 is required for its plasma membrane localization and FA uptake activity[20]. DHHC4 palmitoylates CD36 at Golgi and creates a sorting signal, and DHHC5 maintains CD36 at the cell surface by protecting it from depalmitoylation[20]. It is unknown about whether and how the palmitoylation of CD36 is dynamically regulated.

Endocytosis is an essential way to transport materials, especially large molecules or those hard to penetrate the plasma membrane, into cells[21]. Here, we report that FAs can be transported into cells by CD36-mediated endocytosis. We figure out the signaling pathway and the machineries required for the process, and identify dynamic palmitoylation of CD36 as a key regulatory mechanism. Restricting CD36 either at palmitoylated or depalmitoylated state abolishes its FA uptake activity. Importantly, blocking the endocytosis significantly inhibits CD36-dependent FA uptake, lipid droplet growth, and weight gain on high-fat diet (HFD). We have thus uncovered CD36-mediated endocytosis as a physiologically important pathway to deliver FAs and provided valuable insights on how proteins facilitate the transport of FAs into cells.

## Results

**Long-chain FAs trigger internalization of CD36.** To study changes in subcellular localization of CD36 during FA uptake, we treated 3T3-L1 adipocytes with BSA-conjugated oleate for various time, and performed immunostaining of endogenous CD36 in a 3-dimensional (3D) mode. CD36 stayed on the plasma membrane up to 15 min after oleate treatment. Starting from 30 min, CD36 was internalized and displayed intracellular puncta-like signals inside the cells. By 1 h, CD36 started to wrap around lipid droplets, which became more obvious at 2 and 4 h (Fig. 1a and Supplementary Movies 1–6). As CD36 is primarily localized in

the caveolae structures[8], we have also examined the dynamic localization of Caveolin1 (CAV1), a scaffold protein of caveolae, and found that CAV1 co-migrated with CD36 (Fig. 1a and Supplementary Movies 1–6). To better illustrate the internalization of CD36, we checked the colocalization of CD36 with ATP1A1, a subunit of Na$^+$/K$^+$-ATPase that is usually localized on the plasma membrane. As shown in Supplementary Fig. 1a, ATP1A1 remained on the plasma membrane in either BSA- or oleate-treated cells, whereas CD36 showed clear intracellular localization in oleate-treated cells. To further confirm the results, we examined the caveolae structures by electron microscopy. In BSA-treated cells, caveolaes were mainly localized on the plasma membrane; however, large amounts of caveolaes were internalized 1 h after oleate treatment (Supplementary Fig. 1b). Furthermore, we performed surface biotinylation assay to examine the plasma membrane content of CD36 and ATP1A1. As Triton X-100 used in the assay would not disrupt the caveolae structures, capture of biotinylated CD36 could also pull down CAV1, and we therefore detected CAV1 at the same time. Consistently, the plasma membrane content of ATP1A1 did not change much before and after oleate treatment, but the plasma membrane content of CD36 and CAV1 significantly decreased with time (Fig. 1b and Supplementary Fig. 1c).

Considering that the FA binding pocket of CD36 is on the outer layer of plasma membrane and CAV1 on the inner side, we wondered whether CD36 and CAV1 would depend on each other for the internalization in a way that CD36 acts like a receptor to receive the signal from FAs and CAV1 provides structural support. Indeed, when either one of them was knocked down in 3T3-L1 adipocytes, the other one would no longer get internalized after oleate treatment (Fig. 1c), which is further confirmed by surface biotinylation assay and quantification of surface CD36 content (Fig. 1d and Supplementary Fig. 1e), suggesting that the internalization of them is CD36-mediated caveolar endocytosis.

To confirm that the internalization is triggered by the binding of FAs to CD36, we generated the K164R mutant of CD36 which lacks FA binding activity[22], and electroporated WT CD36 or K164R into the adipocytes derived from stromal vascular fractions (SVFs) isolated from inguinal white adipose tissue (iWAT) of Cd36$^{-/-}$ mice. As expected, WT CD36 was localized on the plasma membrane in BSA-treated cells, and internalized upon oleate treatment. In contrast, K164R did not respond to oleate and stayed on the plasma membrane in both BSA- and oleate-treated cells (Supplementary Fig. 1d).

To test whether all long-chain FAs could trigger the endocytosis, we treated 3T3-L1 adipocytes with FAs possessing different chain lengths or saturation including myristate (C14:0), palmitate (C16:0), stearate (C18:0), linoleate (C18:2), and arachidonate (C20:4), and found that all FAs were able to trigger internalization of CD36 (Fig. 1e). The results were further confirmed by surface biotinylation and quantification of the surface content of CD36 (Fig. 1f and Supplementary Fig. 1f).

We then examined whether CD36 would be recycled to the plasma membrane after withdrawal of FAs from the medium. As shown in Fig. 1g, 30 min after removal of oleate, a small fraction of CD36 started to relocate to the plasma membrane. By 1 h, the majority of CD36 was recycled to the plasma membrane. And by 4 h, almost all CD36 was observed on the plasma membrane. These results were further confirmed by surface biotinylation assay (Fig. 1h and Supplementary Fig. 1g).

**CD36-mediated endocytosis delivers FAs into cells.** As CD36 binds FAs[14], we wondered whether the FA-triggered endocytosis was to deliver FAs into cells. We chose PacFA as an FA analog to

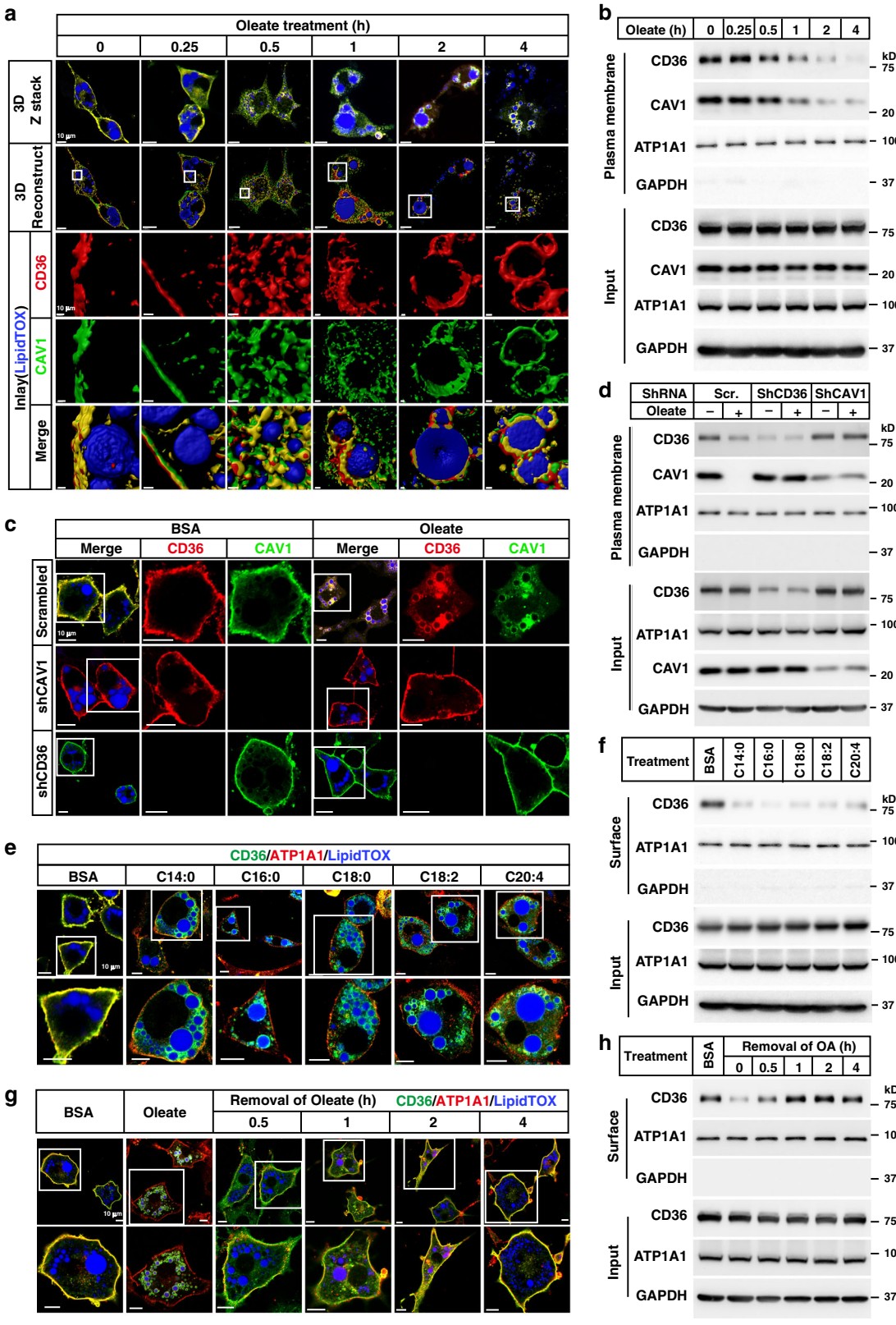

track the movement of FAs in the cells. As illustrated in Fig. 2a, PacFA is a photoactivable and clickable FA analog, and it is able to get incorporated into triglycerides, phospholipids, and cholesterol esters[23]. We treated 3T3-L1 adipocytes with PacFA for 20 min when the endocytosis just started, aiming at trapping the complex of PacFA and CD36 in the endocytosed caveolaes, followed by UV crosslinking, click chemistry with $N_3$-Alexa Fluor

488 to label PacFA, and immunostaining of CD36. Indeed, a clear localization of PacFA and CD36 was observed (Fig. 2a). When quantified, about 56% of PacFA was bound with CD36 inside the cells (Fig. 2b).

To further confirm the results, we pulled down PacFA-bound proteins by using $N_3$-biotin for click chemistry. As shown in Fig. 2c, capture of PacFA by streptavidin beads specifically pulled

**Fig. 1 FAs trigger internalization of CD36. a, b** On day 8 of differentiation, 3T3-L1 adipocytes were pretreated with serum-free medium for 4 h, and then treated with BSA-conjugated oleate (100 μM) for indicated time. **a** One set of cells was subjected to immunostaining with anti-CD36 and anti-CAV1 antibodies. LipidTOX was used to label lipid droplets. Images were taken under a Zeiss LSM-780 microscope in a 3D Z-stack mode and reconstructed using Imaris 9.2.0. **b** The other set of cells was subjected to surface biotinylation assay and blotted with indicated antibodies. **c, d** On day 4 of differentiation, 3T3-L1 cells were infected with lentivirus encoding scrambled shRNA or shRNAs against CD36 or CAV1. On day 5, cells were selected with 5 μg/ml puromycin. On day 8, cells were pretreated as in (**a**) and treated with oleate (100 μM) for 4 h, followed by immunostaining with anti-CD36 and anti-CAV1 antibodies (**c**), or surface biotinylation assay (**d**). **e, f** 3T3-L1 adipocytes were pretreated as in (**a**) and treated with BSA-conjugated FAs with different chain lengths or saturation (100 μM) for 4 h. Cells were subjected to immunostaining with anti-CD36 antibody (**e**), or surface biotinylation assay (**f**). After oleate treatment for 4 h, 3T3-L1 adipocytes were switched to serum-free medium for indicated time and harvested for immunostaining (**g**) and surface biotinylation (**h**). The scale bars were as indicated in each figure. These experiments were repeated at least three times. Source data are provided as a Source Data file.

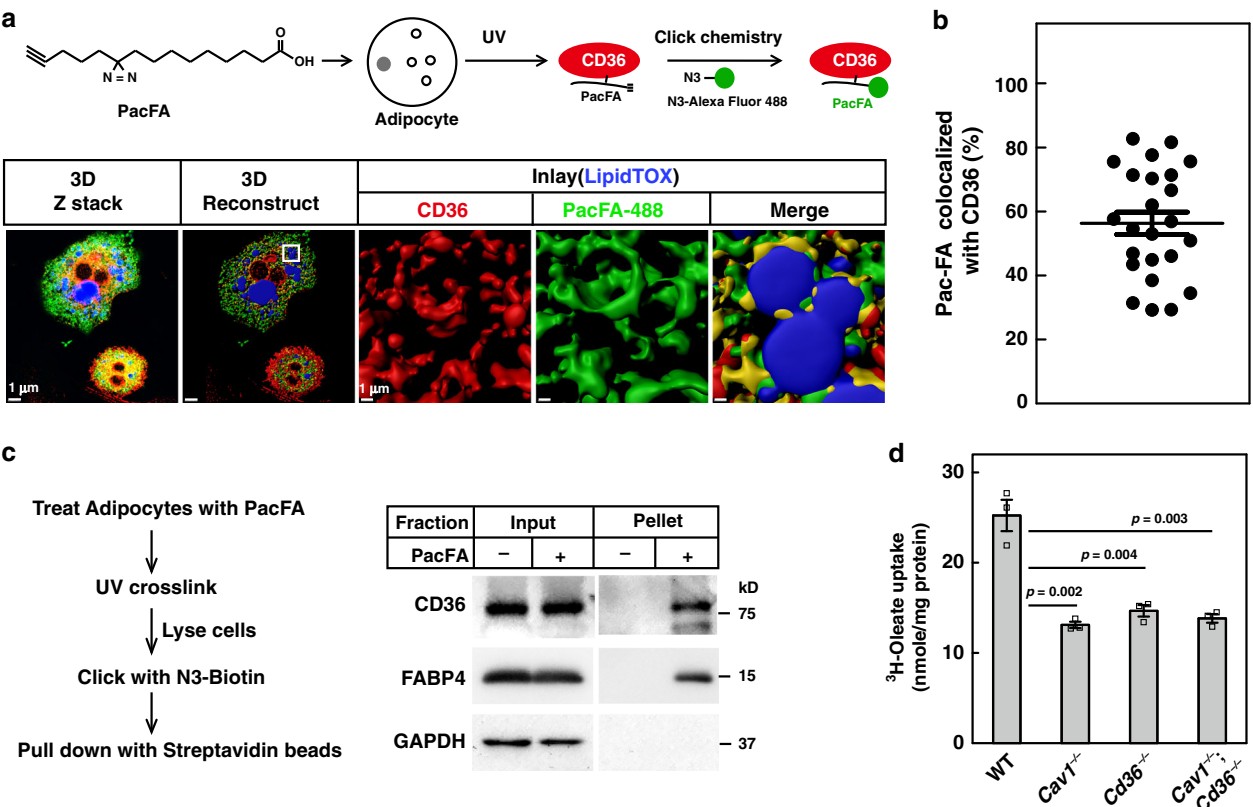

**Fig. 2 CD36-mediated caveolar endocytosis transports FAs into cells. a–c** 3T3-L1 adipocytes were pretreated with serum-free medium for 4 h, and then treated with BSA-conjugated PacFA (50 μM) for 20 min, followed by UV crosslinking on ice for 30 min. **a, b** Cells were subjected to click chemistry using an $N_3$-Alexa Fluro 488, and immunostained with anti-CD36 antibody. Colocalization of PacFA and CD36 was quantified from 24 cells over three independent experiments and plotted in (**b**). The value represents mean ± SEM. **c** Cells were lysed and subjected to click chemistry assay using $N_3$-biotin. PacFA-labeled proteins were captured with streptavidin beads and subjected to western blot using anti-CD36 and anti-FABP4 antibodies. **d** WT, $Cav1^{-/-}$, $Cd36^{-/-}$, and $Cav1^{-/-};Cd36^{-/-}$ SVFs were isolated and differentiated into adipocytes and treated with 100 μM $^3$H-oleate (specific activity, 2268 dpm/ nmol) for 1 h. Lipid fractions were extracted from the cells and subjected to scintillation counting. The radioactive counting was normalized to protein content. Each value represents mean ± SEM obtained from three samples. Two-sided Student's *t* test was performed between WT and each of the knockout cells. These experiments were repeated twice. Source data are provided as a Source Data file.

down CD36. FABP4 was used as a positive control and as expected it was also pulled down by PacFA. These results indicate that CD36 physically delivers FAs inside the cells.

To further confirm that CD36 facilitates FA uptake by endocytosis, we examined whether blocking its endocytosis by depleting CAV1 would abolish its FA uptake. We performed $^3$H-oleate uptake in adipocytes derived from SVFs of $Cav1^{-/-}$, $Cd36^{-/-}$, and $Cav1^{-/-};Cd36^{-/-}$ mice. Indeed, similar to $Cd36^{-/-}$ adipocytes, $Cav1^{-/-}$ adipocytes showed a 40% decrease in $^3$H-oleate uptake, when compared with WT adipocytes (Fig. 2d). Double knockout of $Cav1$ and $Cd36$ did

not further decrease $^3$H-oleate uptake, indicating that caveolar endocytosis is required for CD36-dependent FA uptake.

**Dynamic palmitoylation of CD36 is required for FA uptake.** We then moved on to study how CD36 was dissociated from the plasma membrane. As palmitoylation is required for the plasma membrane localization of CD36[20], we therefore wondered whether CD36 would get depalmitoylated before internalization. We checked changes in the palmitoylation levels of CD36 after various time of oleate treatment. As shown in Fig. 3a, palmitoylation of CD36 started to decrease even 5 min after oleate

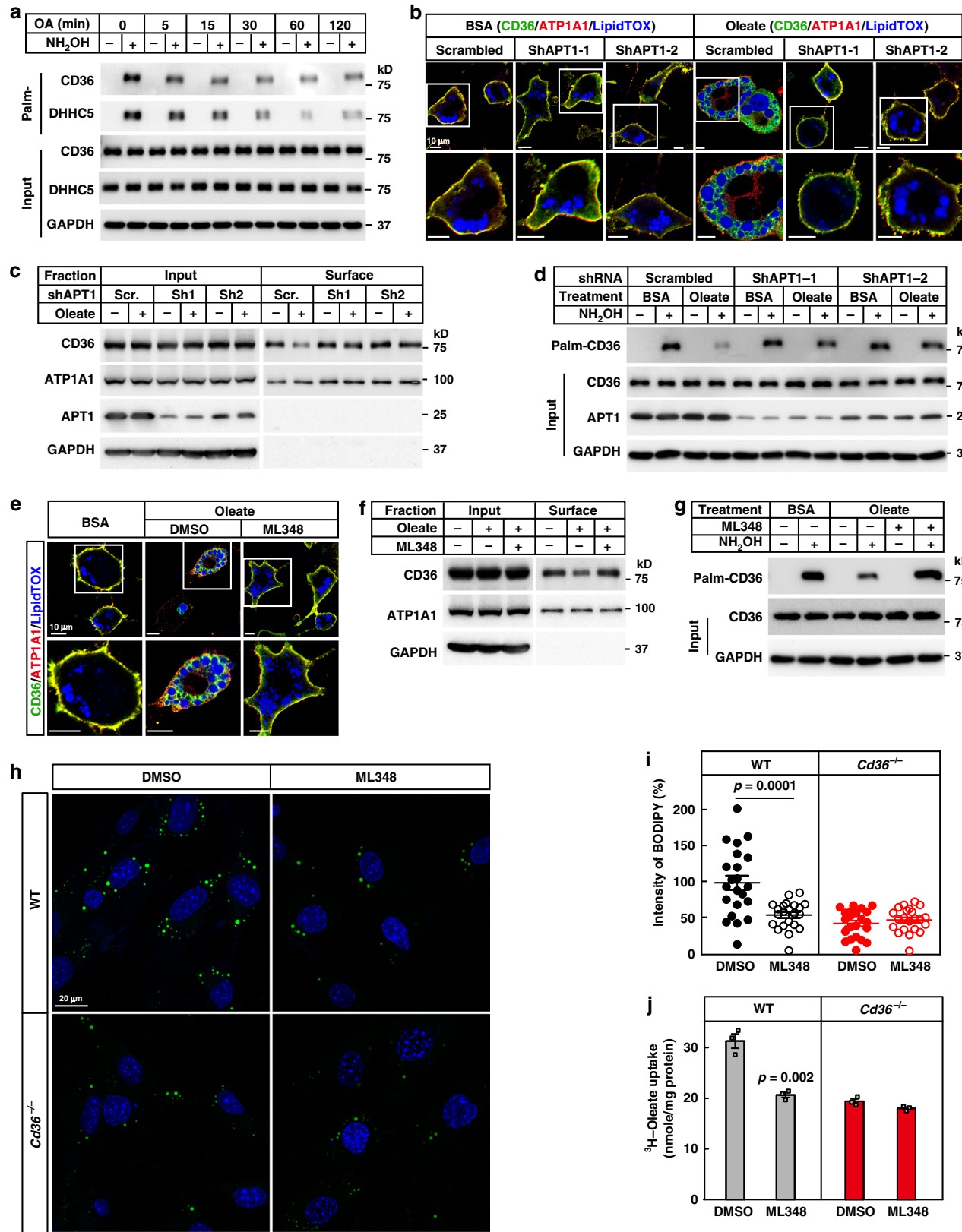

treatment, and reached nadir around 30–60 min. Similar observation was made in the palmitoylation of DHHC5, the enzyme that protects CD36 from depalmitoylation[20], but there was no change in palmitoylation of CAV1 (Supplementary Fig. 2a). Consistently, when oleate was removed from the medium, CD36 and DHHC5 got re-palmitoylated (Supplementary Fig. 2b),

suggesting that CD36 and DHHC5 go through dynamic palmitoylation during the FA-induced endocytosis.

To examine whether depalmitoylation of CD36 is required for the endocytosis, we treated 3T3-L1 adipocytes with palmostatin B (PalmB), a general inhibitor of depalmitoylation, and found that oleate-induced internalization of CD36 was blocked

**Fig. 3 Depalmitoylation of CD36 is required for the endocytosis of FAs. a** 3T3-L1 adipocytes were pretreated and treated with oleate (100 μM) as in Fig. 1a for indicated time. Cells were harvested for Acyl-RAC assay followed by immunoblotting using indicated antibodies. **b–d** Control (scrambled) and APT1 knockdown 3T3-L1 adipocytes were pretreated and treated with BSA or oleate (100 μM) for 1 h. Cells were then harvested for immunostaining (**b**), surface biotinylation (**c**), and Acyl-RAC (**d**) assays. **e–g** 3T3-L1 adipocytes were pretreated and treated with ML348 (10 μM) for 1 h, followed by BSA or oleate treatment for another 1 h. Cells were then harvested for immunostaining (**e**), surface biotinylation (**f**), and Acyl-RAC (**g**) assays. **h, i** On day 0, WT and $Cd36^{-/-}$ SVFs were set up at $4 \times 10^4$ cells per well in a six-well plate. On day 2, cells were pretreated with serum-free medium for 4 h, and treated with ML348 (10 μM) for 1 h, followed by treatment with oleate (100 μM) and BODIPY 493/503 (0.1 μg/ml) for 4 h. **h** Cells were then fixed and imaged on a Zeiss LSM-780 confocal microscope. **i** Quantification of BODIPY fluorescent intensity was performed using ZEN 2010 software. Each value represents mean ± SEM fluorescent intensity per cell from 20 cells. The value in DMSO-treated WT cells was normalized to 1.0. Two-sided Student's $t$ test was performed between DMSO- and ML348-treated cells. DAPI was used to indicate the nuclei. **j** WT and $Cd36^{-/-}$ SVFs were differentiated into adipocytes and pretreated with ML348 (10 μM) for 1 h, followed by treatment with 100 μM $^3$H-oleate (specific activity, 2268 dpm/nmol) for 1 h. Lipid fractions were extracted from the cells and subjected to scintillation counting. The radioactive counting was normalized to protein content. Each value represents mean ± SEM obtained from three samples. Two-sided Student's $t$ test was performed between DMSO- and ML348-treated cells. These experiments were repeated at least twice. Source data are provided as a Source Data file.

(Supplementary Fig. 2c–e). And as expected, PalmB indeed reversed oleate-induced depalmitoylation of CD36 (Supplementary Fig. 2f).

To further confirm the results, we went on to identify the depalmitoylase of CD36. There are mainly five depalmitoylases, including APT1, APT2, and ABHD17A–C[24,25]. We have previously shown that inhibition of the depalmitoylase(s) by PalmB restores plasma membrane localization of CD36 in DHHC5-deficient cells[20]. Taking advantage of this system, we knocked down each of the five depalmitoylases in DHHC5 knockdown 3T3-L1 preadipocytes and checked CD36 localization. As shown in Supplementary Fig. 3a, depletion of APT1, but not the others, restored the plasma membrane localization of CD36. Consistently, knockdown of APT1 restored palmitoylation of CD36 in $DHHC5^{-/-}$ HEK293T cells (Supplementary Fig. 3b). To further confirm the results, we co-expressed each of the five depalmitoylases with CD36 in HEK293T cells, and examined their effects on the palmitoylation of CD36. As shown in Supplementary Fig. 3c–g, only APT1 caused a dramatic decrease in CD36 palmitoylation. Furthermore, overexpression of APT1 dissociated CD36 from the plasma membrane (Supplementary Fig. 3h). These results indicate that APT1 is the depalmitoylase of CD36.

We then knocked down APT1 in 3T3-L1 adipocytes and found that knockdown of APT1 by two separate shRNAs blocked oleate-induced internalization and depalmitoylation of CD36 (Fig. 3b–d and Supplementary Fig. 3i). To further confirm the results, we treated 3T3-L1 adipocytes with ML348, a selective inhibitor of APT1[26]. Similar to knockdown of APT1, inhibition of APT1 by ML348 blocked oleate-induced internalization and depalmitoylation of CD36 (Fig. 3e–g and Supplementary Fig. 3j).

We then examined whether blocking endocytosis by inhibiting APT1 would abolish CD36-mediated endocytic uptake of FAs. We first performed FA uptake in primary preadipocytes isolated from WT and $Cd36^{-/-}$ mice in the presence or absence of ML348. Treatment with ML348 decreased FA uptake in WT cells by 50%, similar to the effect of knocking out $Cd36$, but it did not further decrease FA uptake in $Cd36^{-/-}$ cells (Fig. 3h, i). To further confirm the results, we performed $^3$H-oleate uptake in SVF-derived WT and $Cd36^{-/-}$ adipocytes, and found that ML348 also abolished CD36-dependent $^3$H-oleate uptake, which is about 35% of the activity in WT cells (Fig. 3j).

**DHHC5 is phosphorylated and inactivated during FA uptake.** We next explored what led to depalmitoylation of CD36. The fact that DHHC5 was depalmitoylated after oleate treatment (Fig. 3a) suggested that DHHC5 might get inactivated, as autopalmitoylation of DHHCs is an initial step in catalysis of palmitoylation of

the substrates[27]. As the protein level of DHHC5 did not change after oleate treatment (Fig. 3a), we wondered whether DHHC5 was subjected to a certain posttranslational modification. We pulled down endogenous DHHC5 in BSA- or oleate-treated cells and subjected to mass spectrometry analysis. We examined potential changes in phosphorylation sites of DHHC5, as phosphorylation is usually required for rapid signal transduction. As shown in Fig. 4a, a peptide containing phosphorylated Tyr91 was specifically detected in oleate-treated cells. Topologically, Tyr91 is located in the same cytosolic loop as the DHHC motif (Fig. 4b), and the region flanking this residue is highly conserved in zebrafish, mouse, rat, and human (Fig. 4c), which makes it a potential phosphorylation site to regulate the enzymatic activity of DHHC5.

We then generated a rabbit polyclonal antibody using a synthetic peptide of DHHC5 containing phospho-Tyr91 and studied dynamic phosphorylation of Tyr91 after oleate treatment. As shown in Fig. 4d, phosphorylation of Tyr91 dramatically increased 5 min after oleate treatment, and it dropped to basal level after 30 min. Furthermore, when CD36 was knocked down, oleate-induced phosphorylation of Tyr91 was largely abolished (Fig. 4e), indicating it is a downstream event of FA binding to CD36.

To test the effect of Tyr91 phosphorylation on the enzymatic activity of DHHC5, we generated Y91E and Y91F mutants to mimic the phosphorylated and unphosphorylated DHHC5, respectively. We first examined the enzymatic activity of the mutants in palmitoylating CD36 by reintroducing them into $DHHC5^{-/-}$ HEK293T cells. While Y91F and WT DHHC5 restored palmitoylation of CD36, Y91E could barely do it (Fig. 4f). To further confirm the results, we tested the enzymatic activity the mutants in palmitoylating Flotillin-2, which was previously shown to be a substrate of DHHC5[28]. Consistently, Y91F had similar activity with WT DHHC5 in palmitoylating Flotillin-2, but Y91E showed much decreased activity (Supplementary Fig. 4). These results indicate that phosphorylation of Tyr91 inactivates DHHC5.

We then reintroduced WT, Y91E, or Y91F of DHHC5 into $Dhhc5^{-/-}$ adipocytes by electroporation, and examined CD36 localization in BSA- or oleate-treated cells. As expected, when WT DHHC5 was reintroduced, CD36 was restored to the plasma membrane in BSA-treated cells, and internalized in oleate-treated cells (Fig. 4g). When Y91E was introduced, CD36 exhibited an intracellular localization when the cells were treated with either BSA or oleate, which further confirms that phosphorylation at Tyr91 inactivates DHHC5. When Y91F was introduced, CD36 stayed on the plasma membrane under both conditions, confirming that Y91F is a constitutively active form of DHHC5. The results also indicate that dynamic inactivation of DHHC5 is required for the internalization of CD36.

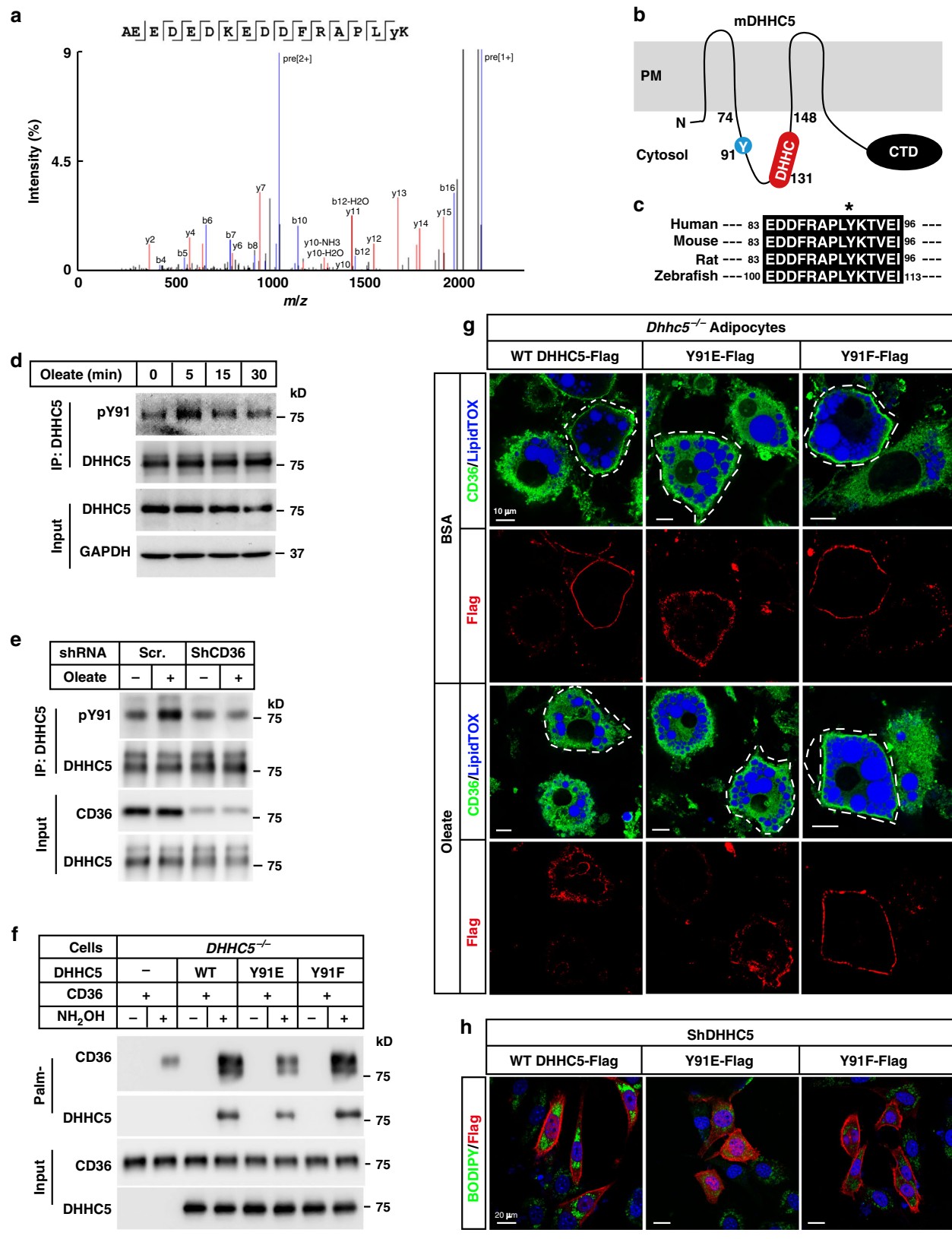

We then examined the effect of Tyr91 phosphorylation on FA uptake. We have previously shown that lack of DHHC5 abolished CD36-dependent FA uptake in 3T3-L1 preadipocytes[20]. We then reintroduced WT, Y91E, or Y91F into DHHC5-deficient preadipocytes and performed FA uptake. In contrast to WT DHHC5, neither Y91E nor Y91F restored the FA uptake activity in DHHC5 knockdown 3T3-L1 preadipocytes (Fig. 4h), indicating that dynamic inactivation of DHHC5 is required for the endocytic uptake of FAs.

**LYN phosphorylates DHHC5 at Tyr91**. We next went on to identify the kinase that phosphorylates DHHC5 at Tyr91.

**Fig. 4 Oleate triggers phosphorylation and inactivation of DHHC5. a** On day 8 of differentiation, 3T3-L1 adipocytes were pretreated with serum-free medium for 4 h, and then treated with BSA or BSA-conjugated oleate (100 µM) for 15 min. Cells were harvested and subjected to immunoprecipitation using anti-DHHC5 antibody. The bands of DHHC5 were cut out and sent for mass spectrometry analysis of the phosphorylation sites. The spectrum of pY91-containing peptide was shown. **b** Indication of Y91 and DHHC motif on a topology of mouse DHHC5, which was predicted at UniProt (www.uniprot.org). **c** Alignment of the Y91 containing regions of DHHC5 proteins from human, mouse, rat, and zebrafish. Control and CD36 knockdown 3T3-L1 adipocytes were pretreated as in (**a**), and treated with oleate (100 µM) for indicated time (**d**) or 5 min (**e**). Cells were harvested, immunoprecipitated with anti-DHHC5 antibody, and immunoblotted with an anti-pY antibody to detect phosphorylation of DHHC5. **f** On day 0, $DHHC5^{-/-}$ HEK293T cells were set up at $7.5 \times 10^5$ cells per 6-cm dish. On day 2, cells were transfected with 0.5 µg CD36-Flag/pCDH-puro and 0.5 µg of indicated DHHC5 WT, Y91E, or Y91F/pCDH-puro. On day 3, cells were harvested for Acyl-RAC assay and blotted with indicated antibodies. **g** SVFs were isolated from $Rosa\text{-}Cre^{ERT2};Dhhc5^{f/f}$ mice, treated with 4-OHT to induce deletion of $Dhhc5$, and subjected to differentiation. On day 6 of differentiation, cells were electroporated with WT, Y91E, or Y91F of DHHC5-Flag. On day 8, cells pretreated and treated with oleate as in Fig. 1c. Cells were subjected to immunofluorescence using anti-CD36 and anti-Flag antibodies. LipidTOX was used to indicate lipid droplets. Cells expressing WT, Y91E, or Y91F of DHHC5 were outlined as indicated. Scale bar, 10 µm. **h** DHHC5 knockdown 3T3-L1 preadipocytes were transfected with WT, Y91E, or Y91F of DHHC5-Flag. Cells were treated with oleate (100 µM) and BODIPY 493/503 (0.1 µg/ml) for 8 h, followed by immunostaining with anti-Flag M2 antibody. DAPI was used to indicate the nuclei. These experiments were repeated at least twice. Source data are provided as a Source Data file.

Bioinformatic analysis using GPS 3.0 (http://gps.biocuckoo.org) revealed that Tyr91 might be a potential substrate of SFKs. To test the possibility, we treated adipocytes with PP2, a selective inhibitor of SFKs[29], and found that oleate-induced Tyr91 phosphorylation of DHHC5 was largely suppressed (Fig. 5a). Furthermore, oleate-induced endocytosis of CD36 and depalmitoylation of CD36 and DHHC5 were also inhibited by PP2 (Supplementary Fig. 5a–d).

To search for the kinase, we knocked down each of the four relatively abundant SFKs (LYN, FYN, SRC, and YES1) in 3T3-L1 adipocytes (Supplementary Fig. 5e), and examined the effect on oleate-induced Tyr91 phosphorylation of DHHC5. While knockdown of FYN, SRC, or YES1 showed no effect, knockdown of LYN suppressed oleate-induced phosphorylation of DHHC5 at Tyr91 (Fig. 5b and Supplementary Fig. 5f). And two separate shRNAs showed similar effects (Supplementary Fig. 5g). When co-expressed in HEK293T cells, LYN phosphorylated WT DHHC5, but not Y91E or Y91F (Fig. 5c). Furthermore, knockdown of LYN with two separate shRNAs blocked oleate-induced internalization of CD36 and depalmitoylation of CD36 and DHHC5 (Fig. 5d–f and Supplementary Fig. 5h), confirming that LYN phosphorylates DHHC5 at Tyr91.

We then examined LYN activation after oleate treatment. Phosphorylation of LYN at Tyr396, which indicates the activation of LYN, was significantly increased 5 min after oleate treatment, and the signal dropped to basal level at 15 min (Fig. 5g). Furthermore, when CD36 was knocked down, oleate-induced phosphorylation at Tyr396 was largely abolished (Fig. 5h), indicating that LYN is a downstream kinase of CD36.

To further confirm the requirement of LYN in CD36-mediated endocytosis of FAs, we inhibited LYN with PP2 to block CD36 internalization and performed FA uptake. Similar to the effect of ML348, treatment with PP2 abolished CD36-dependent FA uptake in both preadipocytes and adipocytes (Fig. 5i–k).

**Depalmitoylated CD36 recruits SYK to facilitate FA uptake.** We then sought to explore why depalmitoylation of CD36 is required for the endocytosis. We first examined the recruitment of dynamin, VAV, and JNK, which were previously reported to participate in caveolar endocytosis or phagocytosis of oxLDL by CD36 in macrophages[30–34]. We treated 3T3-L1 adipocytes with the inhibitors of dynamin (Dyngo4a), VAVs (Azathioprine), JNK (SP600125), or LYN (PP2, as a positive control), and found that all inhibitors blocked oleate-induced endocytosis of CD36 (Fig. 6a, b and Supplementary Fig. 6a).

To examine whether depalmitoylation of CD36 happened before the recruitment of these factors above, we treated 3T3-L1 adipocytes with ML348 to block depalmitoylation of CD36 and

examined the activation of VAV and JNK by checking their phosphorylation. As shown in Fig. 6c, oleate treatment caused increased phosphorylation of VAV (pTyr174) and JNK (pThr183/Tyr185), but such effect was largely blocked when cells were pretreated with ML348, indicating that depalmitoylation of CD36 is required for the recruitment of the machineries. These results also raised a possibility that a certain kinase might be recruited by depalmitoylated CD36 to phosphorylate VAV and JNK.

We then went on to search for the kinase. We isolated depalmitoylated CD36 by expressing CD36-Flag in $DHHC5^{-/-}$ HEK293T cells and treating the cells further with 2-bromopalmitate, an inhibitor of DHHCs, to block palmitoylation of CD36. We then incubated depalmitoylated CD36-Flag with tissue lysates of gWAT prepared from mice fasted and refed with HFD, and the potential interacting proteins of depalmitoylated CD36 were analyzed by silver staining and mass spectrometry (Fig. 6d). Among the proteins, SYK, a tyrosine kinase, caused our attention, as it has been shown to be usually activated after SFKs[35] and required for CD36-mediated phagocytosis of oxLDL in macrophages[36]. We found that oleate treatment enhanced the interaction between CD36 and SYK (Supplementary Fig. 6b). Furthermore, SYK was activated after oleate treatment, as indicated by the phosphorylation at Tyr525/526 (Supplementary Fig. 6c), and recruited to the plasma membrane (Supplementary Fig. 6d). Blockage of CD36 depalmitoylation by ML348 abolished oleate-induced phosphorylation of SYK (Fig. 6e), indicating that SYK is activated after depalmitoylation of CD36.

To examine the requirement of SYK in the endocytosis, we then treated 3T3-L1 adipocytes with piceatannol, a SYK inhibitor, and found that it largely abolished oleate-induced phosphorylation of VAV and JNK (Fig. 6f), and internalization of CD36 (Fig. 6g, h and Supplementary Fig. 6e). And piceatannol did not affect oleate-induced depalmitoylation of CD36 (Fig. 6i), confirming that SYK activation is a downstream event of CD36 depalmitoylation. Furthermore, knocking down SYK in 3T3-L1 adipocytes by two separate shRNAs blocked oleate-induced internalization of CD36 (Supplementary Fig. 6f–i).

To examine whether blocking endocytosis by inhibiting SYK would also abolish CD36-dependent FA uptake, we performed FA uptake in the presence or absence of piceatannol. Very similar to the inhibition of APT1 or LYN, piceatannol treatment also abolished CD36-dependent FA uptake in both preadipocytes and adipocytes (Fig. 6j–l).

**Blocking endocytosis inhibits physiological function of CD36.** We then went on to evaluate the physiological significance of CD36-mediated endocytosis of FAs. As a major physiological

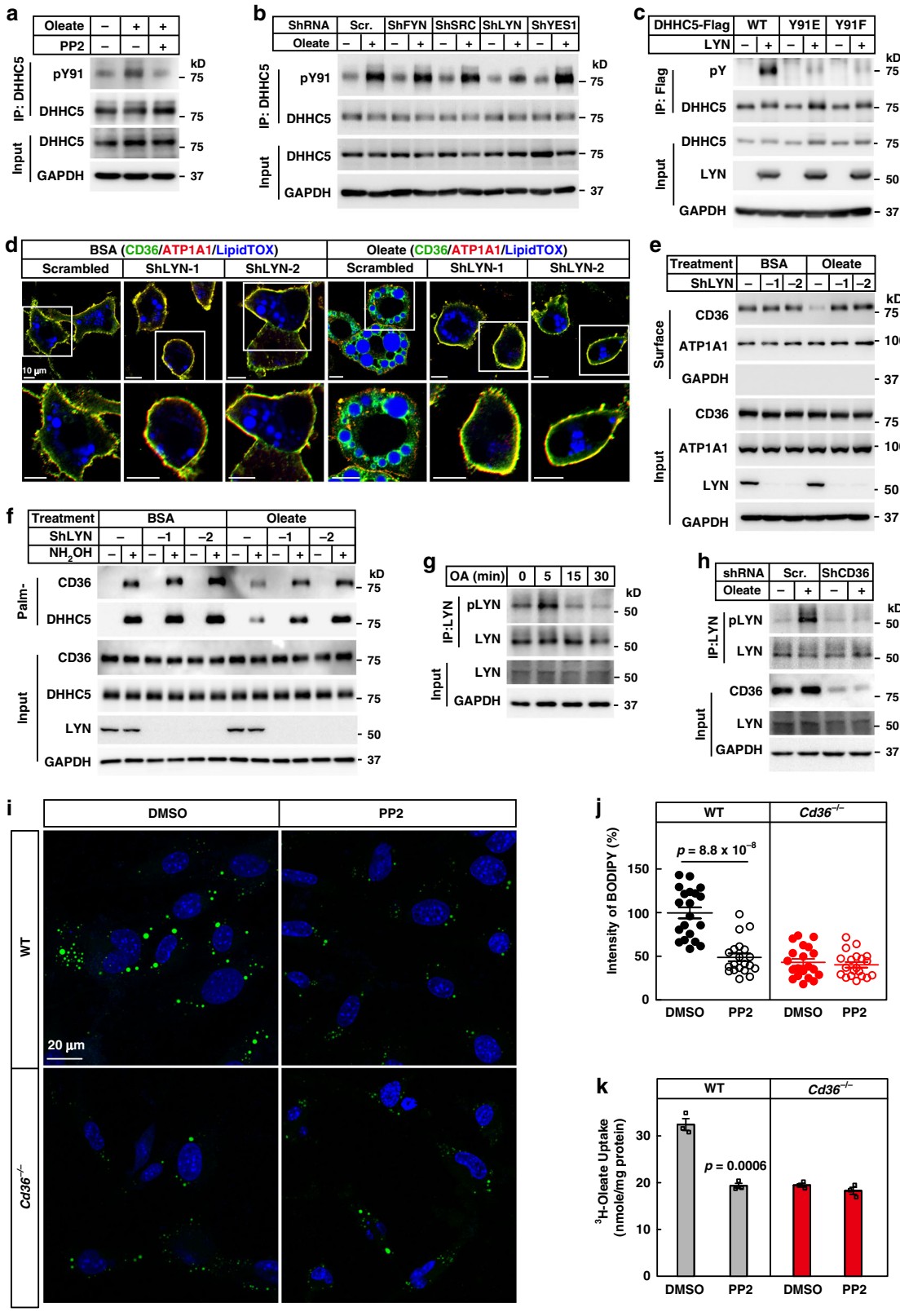

function of FA uptake is to store FAs in lipid droplets of adipocytes, we explored the effect of blocking the endocytosis on lipid droplet growth. We pretreated WT and $Cd36^{-/-}$ adipocytes with PP2, ML348, or piceatannol to inhibit LYN, APT1, or SYK, and monitored oleate-induced growth of lipid droplets using live imaging microscopy. In DMSO-treated WT adipocytes, oleate treatment promoted the growth of lipid droplets with time, and

by 105 min, the volume of lipid droplets per cell doubled (Fig. 7a and Supplementary Fig. 7). However, such effect of oleate was significantly suppressed when WT cells were pretreated with either PP2, ML348, or piceatannol. In DMSO-treated $Cd36^{-/-}$ adipocytes, there was very little or no obvious growth of lipid droplets during the 2-h period. Treatment with any of the inhibitors did not further decrease lipid droplet growth (Fig. 7a and

**Fig. 5 LYN phosphorylates DHHC5 and is required for the endocytosis of FAs. a** 3T3-L1 adipocytes were pretreated with serum-free medium for 4 h and PP2 (15 μM) for 1 h, followed by treatment with BSA or oleate (100 μM) for 5 min. Cells were harvested and immunoprecipitated with anti-DHHC5 antibody to detect phosphorylation of Tyr91. **b** 3T3-L1 adipocytes transduced with indicated shRNA were set up and treated with oleate for 5 min. Cells were harvested to detect Tyr91 phosphorylation as in (**a**). **c** On day 0, HEK293T cells were set up at $2.5 \times 10^5$ cells per 6-cm dish. On day 2, cells were transfected with 0.5 μg of WT, Y91E, or Y91F of DHHC5-Flag/pCDH-puro and/or 0.5 μg LYN/pCDNA3.3. On day 3, cells were harvested and immunoprecipitated with anti-Flag M2 beads to detect phosphorylation of DHHC5 (pY). Control and LYN knockdown adipocytes were pretreated and treated with BSA or oleate (100 μM) for 1 h, followed by immunostaining (**d**), surface biotinylation (**e**), and Acyl-RAC (**f**) assays. Control and CD36 knockdown 3T3-L1 adipocytes were pretreated and treated with oleate (100 μM) for indicated time (**g**) or 5 min (**h**). Cells were harvested and subjected to immunoprecipitation of LYN to detect phosphorylation of LYN (Y396). **i, j** WT and $Cd36^{-/-}$ SVFs were set up and subjected to FA uptake as in Fig. 3h, i, except that cells were pretreated with PP2 (15 μM). Each value represents mean ± SEM obtained from 20 cells. Two-sided Student's t test was performed between DMSO and PP2 treated cells. **k** WT and $Cd36^{-/-}$ adipocytes were set up and subjected to $^3$H-oleate uptake as in Fig. 3j, except that cells were pretreated with PP2 (15 μM). Each value represents mean ± SEM obtained from three samples. Two-sided Student's t test was performed between DMSO and PP2 treated cells. These experiments were repeated at least twice. Source data are provided as a Source Data file.

Supplementary Fig. 7), indicating that blocking endocytosis abolished CD36-dependent lipid droplet growth in adipocytes.

We then sought to explore the physiological function of CD36-mediated endocytosis of FAs on diet-induced obesity in mice. To block the endocytosis in mice, we chose bafetinib, a selective dual Bcr-ABL/LYN inhibitor[37], and entospletinib, a selective SYK inhibitor[38]. As expected, both compounds inhibited oleate-induced endocytosis of CD36 in 3T3-L1 adipocytes (Supplementary Fig. 8a, b). In WT mice, treatment with bafetinib or entospletinib significantly reduced weight gain on HFD (Fig. 7b). Vehicle-treated $Cd36^{-/-}$ mice gained less weight than vehicle-treated WT mice, but treatment with either compound did not further decrease the body weights. We have also performed glucose tolerance test on week 8. In WT mice, bafetinib- or entospletinib-treated mice cleared glucose much faster than vehicle-treated mice, and such effect was CD36 dependent (Supplementary Fig. 9a). We monitored food intake in the mice and found that neither bafetinib nor entospletinib changed the food intake in WT or $Cd36^{-/-}$ mice (Supplementary Fig. 9b), suggesting that the decrease in body weights of WT mice was not due to suppression of appetite. At the end of the experiment, we dissected the mice and found that treatment with bafetinib or entospletinib significantly reduced the fat mass of iWAT and gWAT, and the sizes of adipocytes in WT, but not $Cd36^{-/-}$ mice (Fig. 7c, d and Supplementary Fig. 9c). To explore where the FAs were going to in bafetinib- or entospletinib-treated WT mice, we measured plasma levels of free FAs, and found that neither compound raised the plasma free FAs (Supplementary Fig. 9d). Similar observations were made in plasma triglycerides (Supplementary Fig. 9e). We then measured liver triglyceride levels and found that both compounds slightly but significantly raised liver triglyceride levels in WT mice, similar to those in the $Cd36^{-/-}$ mice (Supplementary Fig. 9f), suggesting that the decreased FA uptake activity in adipose tissues caused ectopic storage of FAs in liver.

## Discussion

FAs are essential nutrients for cellular functions[1,4], but how they are transported into cells remains unclear. Here, we demonstrate that long-chain FAs trigger CD36-mediated caveolar endocytosis. Using PacFA as an FA analog, we prove that FA-induced internalization of CD36 carries FAs into cells. We have figured out the key machineries and signaling pathway of the endocytic process, and shown that blocking endocytosis by depleting CAV1 or inhibiting APT1, LYN, or SYK all abolishes CD36-dependent FA uptake. Importantly, we show that the endocytic uptake of FAs is required for CD36-dependent lipid droplet growth in adipocytes and HFD-induced weight gain in mice.

It actually makes sense that CD36-mediated caveolar endocytosis represents an effective was to transport FAs into adipocytes.

First, from the point of cellular structures, caveolaes can make up to approximately one-third of the plasma membrane in adipocytes[39,40]. CD36 is also highly abundant in adipocytes[14], and it is primarily localized in the caveolae structures with its FA binding pocket is on the cell surface to capture FAs[8,9]. Such an organization ensures a large scale of caveolar endocytosis during FA uptake. Second, we show that CD36 delivers FAs to lipid droplets. Previous studies have shown that triglyceride synthesis enzymes, GPAT4, AGPAT3, and DGAT2, can relocate from ER to lipid droplets upon treatment with FAs[41–43]. Therefore, CD36-mediated caveolar endocytosis might efficiently couples FA uptake, esterification, and triglyceride synthesis. Third, given the hydrophobic property of FAs, it might be energetically efficient to transport FAs by endocytosis, which has been adopted by other hydrophobic lipid molecules, such as LDL[44], oxLDL[45], and cholesterol[46,47].

Notably, we have identified a central role of dynamic palmitoylation of CD36 in regulating the endocytosis of FAs. CD36 gets depalmitoylated by treatment with FAs, and it gets re-palmitoylated after withdrawal of FAs. While palmitoylated CD36 captures FAs on the cell surface, depalmitoylated CD36 initiates the endocytic process and delivers FAs into cells. Blockage of the depalmitoylation of CD36 by depleting or inhibiting LYN or APT1 abolishes the endocytic uptake of FAs. Furthermore, restricting CD36 at palmitoylated state in Y91F DHHC5-expressing cells or depalmitoylation state in Y91E DHHC5-expressing cells abolishes the FA uptake activity of CD36. Taken together, it is the dynamic palmitoylation that ensures the FA uptake activity of CD36.

Furthermore, we have identified dynamic phosphorylation and inactivation of DHHC5 as the underlying mechanism of dynamic palmitoylation of CD36. We show that binding of FAs to CD36 activates its downstream kinase LYN, thereby converting the extracellular stimulus of FAs into intracellular signaling pathway. LYN then phosphorylates DHHC5 at Tyr91 and inactivates DHHC5, resulting in depalmitoylation of CD36 by APT1. Although there are more than 20 DHHCs in mammals[17,48], how their enzymatic activities are regulated is barely known. Therefore, the dynamic phosphorylation by SFKs might provide a regulatory mechanism for the enzymatic activities of DHHCs family. As SFKs are widely involved in various physiological conditions[49], our findings might expand the physiological functions of DHHCs to these events.

Up to date, several proteins have been reported to facilitate FA uptake, including CD36, CAV1, FATP, and FABPm[7]. Besides CD36, CAV1 has also been reported to get internalized after FA treatment[50,51], though the physiological role was unknown. Our findings here uncover that the internalization of CD36 and CAV1 is to transport FAs into cells by CD36-mediated caveolar endocytosis. It would be interesting to see whether these other proteins would also transport FAs via endocytosis.

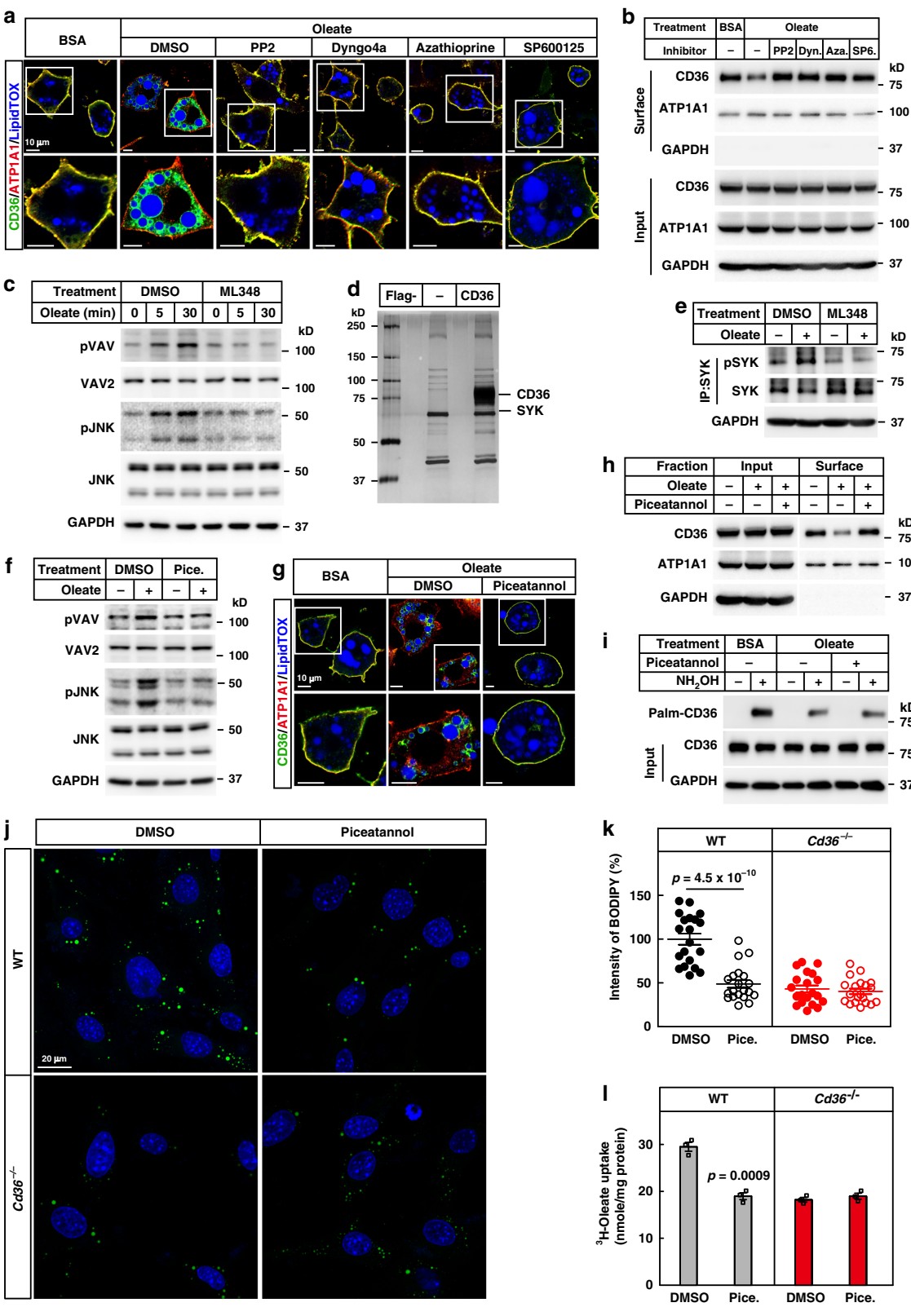

To summarize our findings, we propose a working model of the CD36-mediated endocytosis of FAs (Fig. 8). When FAs bind to CD36, its downstream kinase LYN is activated, and it phosphorylates DHHC5 at Tyr91 and inactivates DHHC5, resulting in subsequent depalmitoylation of CD36 by APT1. The depalmitoylated CD36 then recruits another tyrosine kinase, SYK, which phosphorylates VAV and JNK. VAV function as an adapter of dynamin[32,33], which facilitates the pinching off of caveolaes from the plasma membrane[21]. The activated JNK plays an important role in regulating cytoskeleton re-organization and vesicle transport[52], two key events caveolar endocytosis. Eventually, caveolaes are internalized and deliver FAs to ER for esterification and further distribution to lipid droplets for storage. After delivery, CD36 gets re-palmitoylated

**Fig. 6 Depalmitoylated CD36 recruits SYK to facilitate the endocytosis. a, b** 3T3-L1 adipocytes were pretreated with serum-free medium for 4 h, and DMSO, PP2 (20 μM), Dyngo4a (2 μM), azathioprine (10 μM), or SP600125 (20 μM) for 1 h, followed by treatment with BSA or oleate (100 μM) for 1 h. Cells were harvested for immunostaining (**a**) and surface biotinylation (**b**) assays. **c** 3T3-L1 adipocytes were pretreated with ML348 (10 μM) for 1 h, following by oleate treatment for 0, 5, or 30 min. Cells were harvested and immunoblotted with indicated antibodies. **d** The interacting proteins of depalmitoylated CD36 were isolated as described in "Methods." The eluted fractions were separated on SDS-PAGE, and subjected to silver staining and mass spectrometry. **e** 3T3-L1 adipocytes were pretreated with ML348, followed by BSA or oleate (100 μM) for 5 min. Cells were harvested, subjected to immunoprecipitation with anti-SYK antibody, and blotted with anti-SYK and anti-pSYK (Y525/526). **f** 3T3-L1 adipocytes were pretreated with serum-free medium for 4 h and piceatannol (Pice., 40 μM) for 1 h, followed by treatment with BSA or BSA-conjugated oleate (100 μM) for 30 min. Cells were harvested for immunoblotting with indicated antibodies. **g–i** 3T3-L1 adipocytes were pretreated with serum-free medium for 4 h and piceatannol (40 μM) for 1 h, followed by treatment with BSA or BSA-conjugated oleate (100 μM) for 1 h. Cells were harvested for immunostaining (**g**), surface biotinylation (**h**), and Acyl-RAC (**i**) assays. **j, k** WT and $Cd36^{-/-}$ SVFs were set up and subjected to FA uptake as in Fig. 3h, i, except that cells were pretreated with piceatannol (40 μM). Each value represents mean ± SEM obtained from 20 cells. Two-sided Student's $t$ test was performed between DMSO and piceatannol-treated cells. **l** WT and $Cd36^{-/-}$ adipocytes were set up and subjected to $^3$H-oleate uptake as in Fig. 3j, except that cells were pretreated with piceatannol (40 μM). Each value represents mean ± SEM obtained from three samples. Two-sided Student's $t$ test was performed between DMSO and piceatannol-treated cells. The scale bars were as indicated. These experiments were repeated at least twice. Source data are provided as a Source Data file.

and recycled to the plasma membrane for another round of delivery.

## Methods

**Experimental models and subject details.** $Cd36^{-/-}$ mice[53] were purchased from Jackson laboratory, and $Cav1^{-/-}$ mice were purchased from Cyagen Bioscience (KOCMP-21072-Cav1). All mice were housed in colony cages at 25 °C with 12-h light/12-h dark cycles, with 20–60% humidity. The dark cycle began at 7 p.m. All animal studies were performed with the approval of the Institutional Animal Care and Research Advisory Committee at Xiamen University. All mice used in the paper are male on a C57BL6 background.

**Stock preparation.** Stock solutions of BSA-conjugated FAs or PacFA[23] (Avanti Polar Lipids, 900401P) were prepared in 0.15 M NaCl containing 10% (wt/vol) BSA (essentially FA free) at final concentrations of 5 or 10 mM[54]. A stock solution of hydroxylamine HCl (2 M, Sigma-Aldrich, 159417) was freshly prepared in water and was adjusted to pH7.5 with NaOH. A stock solution of azathioprine (Sigma-Aldrich, A4638) was prepared in 0.1 M NaOH at a final concentration of 5 mM. Stock solutions of PalmB (20 mM, Merck, 178501), PP2 (20 mM, MCE, HY-13805), ML348 (10 mM, MCE, HY-100736), SP600125 (20 mM, ApexBIO, A4604), Dyngo4a (2 mM, Abcam, ab120689), and piceatannol (40 mM, MCE, HY-13518) were made up in DMSO. Bafetinib (Efebio, E074675) and entospletinib (Efebio, E012515) were prepared in 0.5% methyl cellulose in saline right before the oral gavage.

**Plasmids.** Full length cDNA of mouse *Lyn* was cloned from a cDNA library prepared from testis of a C57BL6 mouse. Human *CD36* and *DHHC5* were cloned in our previous studies[20]. These genes were cloned into either pcDNA3.3, or pCDH-EF1-MCS-IRES-Puro (System Biosciences). For knockdown, two shRNAs of each gene were designed and cloned into pLKO.1 (Addgene, 10878)[55]. The primer sequences are listed in Supplementary Table 1.

**Differentiation of 3T3-L1 preadipocytes.** 3T3-L1 preadipocytes (ATCC) were cultured in DMEM (1 g/L glucose) supplemented with 10% (v/v) NCS (Thermal Fischer Scientific), 100 U/ml penicillin, and 100 mg/ml streptomycin at 37 °C in an atmosphere of 8.8% CO$_2$. Two days after confluence, 3T3-L1 cells were induced by standard hormone cocktails to differentiate into mature adipocytes. Briefly, cells were induced with DMEM containing 10% FCS, 5 μg/ml insulin, 0.5 mM IBMX, and 1 μM dexamethasone for 2 days, treated with DMEM containing 5 μg/ml insulin for 4 days, and switched to DMEM containing 10% FCS for 2 more days. On day 8 of differentiation, mature adipocytes were harvested for further experiments.

**Isolation of stromal vascular fractions and differentiation into adipocytes.** SVFs were isolated from iWATs of 6–8-week-old male WT, $Cd36^{-/-}$, or $Rosa$-$Cre^{ERT2}$;$Dhhc5^{f/f}$ mice. SVFs were cultured and differentiated into mature adipocytes as 3T3-L1 cells.

**Lentivirus production and infection.** HEK293T cells were cultured in DMEM (4.5 g/L glucose) supplemented with 10% (v/v) FCS (Thermo Fisher Scientific), 100 U/ml penicillin, and 100 mg/ml streptomycin at 37 °C in an atmosphere of 5% CO$_2$. For lentivirus packaging, gene overexpressing or shRNA vector was co-transfected with psPAX2 and pMD2.G into HEK293T cells. Medium containing lentiviral particles were either concentrated at 70,000 × g for 2 h or directly aliquoted and stored at −80 °C until use.

For lentiviral infections of 3T3-L1 adipocytes, 3T3-L1 cells, on day 4 of differentiation, were infected with lentivirus encoding scrambled shRNA or shRNA targeting a certain gene in medium containing 8–10 μg/ml polybrene. Cells were selected against 5 μg/ml puromycin on day 5 for at least 48 h before harvest for the described experiments.

**Electroporation of adipocytes.** On day 6 of differentiation, indicated adipocytes in the figure legends were trypsinized and resuspended in electroporation buffer (Lonza-Amaxa). Cells were aliquoted into cuvettes (2 × 10$^6$ cells) and mixed with 2 μg of indicated plasmid in the figure legends. Electroporation was performed using the A-033 program of Nucleofector$^{TM}$ 2b (Lonza-Amaxa). After electroporation, cells were plated to 35-mm dishes for further experiments.

**Protein palmitoylation analysis by Acyl-RAC assay.** Palmitoylated proteins were isolated using resin-assisted capture of S-acylated proteins (Acyl-RAC)[56] with more stringent washing conditions[20]. Briefly, free thiol groups were blocked with 0.1% s-methyl methanethiosulfate at 42 °C for 15 min. Proteins were precipitated by cold acetone at −20 °C for 1 h, washed twice by 70% acetone, and resuspended in 0.6 ml Buffer A (100 mM Hepes, pH7.5, 1 mM EDTA, 1% SDS). Samples were mixed with 20 μl thiopropyl sepharose 6B, and incubated with 0.21 M NH$_2$OH or NaCl with constant rotation at room temperature for 3 h. Resins were washed with Buffer A containing 8 M urea for five times (5 min each) and eluted with 60 μl Buffer A containing 50 mM DTT at room temperature for 20 min. Eluted fractions from Acyl-RAC assay were analyzed by sliver staining, mass spectrometry, or western blot. Blots were developed in a ChemStudio imaging system (Analytik Jena AG).

**Immunoprecipitation and western blot.** Fully differentiated 3T3-L1 adipocytes were lysed in Buffer B (50 mM Tris (pH7.4), 150 mM NaCl, 1% Triton, and 0.1% SDS) containing protease and phosphatase inhibitors (MCE). Cell lysate was immunoprecipitated with anti-DHHC5 (Sigma, HPA014670), anti-CD36 (Novus, NBP2-54790), anti-SYK (CST, 13198s), or anti-LYN (CST, 2796s) and pulled down with protein A/G beads. Pellet was washed five times with Buffer C (50 mM Tris (pH7.4), 150 mM NaCl and 0.1% Triton), and incubated with SDS loading buffer at 37 °C for 1 h.

To generate pTyr91 antibody of DHHC5, a synthetic phospho-peptide CDKEDDFRAPL(pY)KTVE was conjugated to KLH and used to immunize rabbits. After three immunizations, anti-serum was affinity purified and used for western blot.

The following antibodies were used for western blot: anti-Flag (1:1000, Sigma-Aldrich, F1804), anti-HA (1:5000, Roche, 11867423001), anti-FABP4 (1:1000, Abclonal, A11481), anti-CD36 (1:1000, Sino Biological Inc., 80263-T48), anti-CAV1 (1:1000, CST, 3238s), anti-DHHC5 (1:1000, Sigma-Aldrich, HPA014670), anti-pTyr 4G10 (1:1000, EMD Millipore, 05-321), anti-LYN (1:1000, CST, 2796s), anti-pLYN (Y396) (1:1000, Abcam, 226778), anti-JNK (1:1000, CST, 9252s), anti-pJNK (T183/Y185) (1:1000, CST, 9255s), anti-VAV2 (1:1000, Proteintech, 60211-1-Ig), anti-pVAV(Tyr174) (1:1000, Santa Cruz, SC-135788), anti-SYK (1:1000, CST, 13198s), anti-pSYK (Y525/526) (1:1000, CST, 2710s), and anti-GAPDH (1:1000, Proteintech, 60004-1-Ig).

Blots were developed and band intensities were quantified using VisionWorks software on a ChemStudio imaging system (Analytik Jena AG). Uncropped blots are available in the Source Data file.

**Identification of interacting proteins of depalmitoylated CD36.** On day 0, $DHHC5^{-/-}$ HEK293T cells were set up 7.5 × 10$^5$ cells per 10-cm dish (five dishes). On day 2, cells were transfected with 6 μg CD36-Flag/pCDH-puro. On day 3, cells were treated with 2-bromopalmitate (100 μM) for 24 h. On day 4, cells were

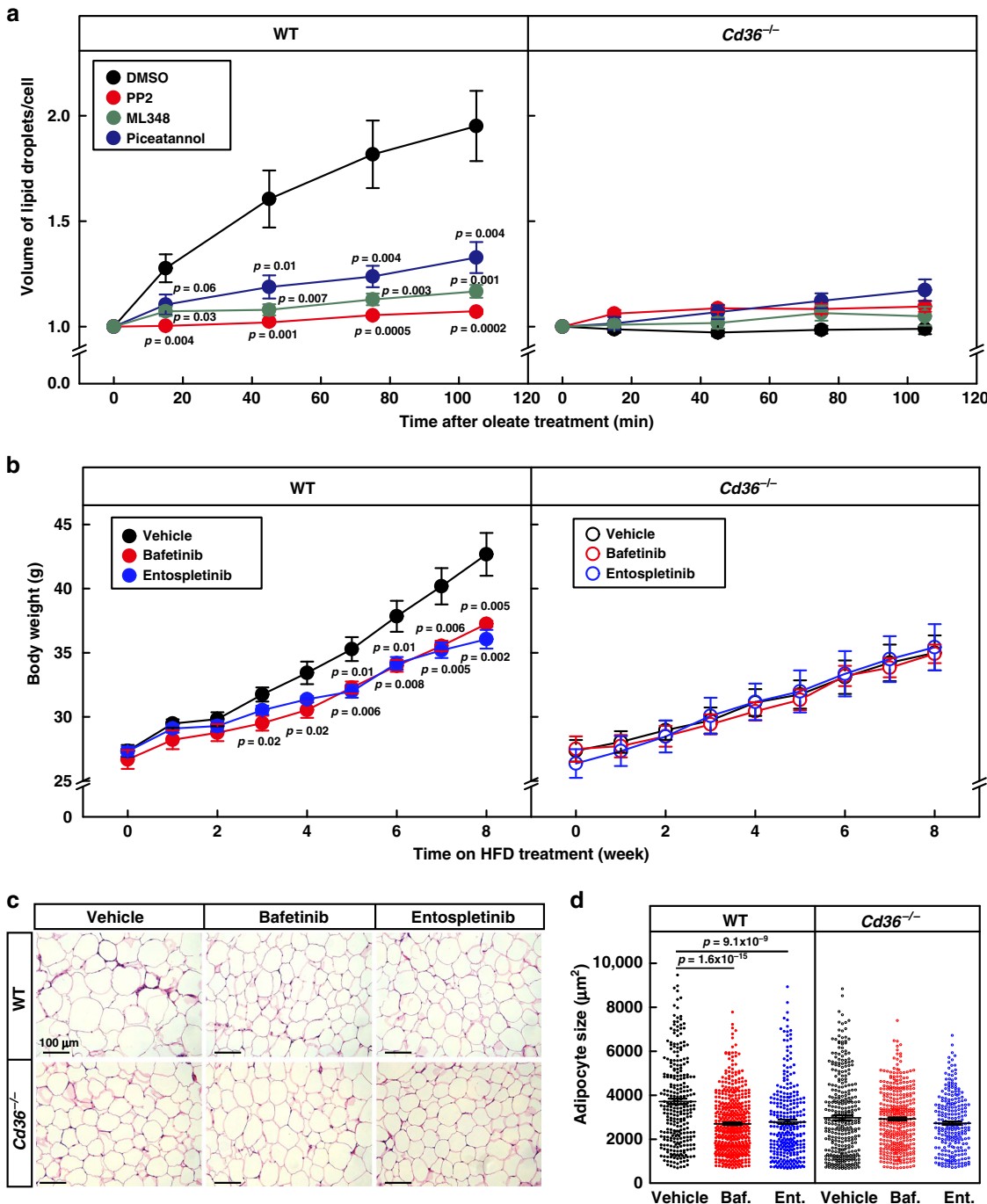

**Fig. 7 Blocking endocytosis inhibits lipid droplet growth and HFD-induced obesity. a** On the day of experiment, WT and $Cd36^{-/-}$ adipocytes were pretreated with PP2 (20 μM), ML348 (10 μM), or piceatannol (40 μM) for 1 h, then treated with oleate (100 μM) and BODIPY 493/503 (0.1 μg/ml), and subjected to live imaging on a Zeiss LSM-780 confocal microscope in a 3D Z-stack mode for 2 h. The number of cells was 12, 9, 8, and 10 for DMSO, PP2, ML348, and piceatannol-treated WT cells, and 8, 8, 8, and 10 for corresponding groups in $Cd36^{-/-}$ cells, respectively. Total volume of lipid droplets in each cell was calculated. Each value represents mean ± SEM, and the volume at 0 min was normalized to 1.0. Two-sided Student's *t* test was performed between DMSO- and PP2-, ML348- or piceatannol-treated cells. **b**–**d** WT and $Cd36^{-/-}$ mice (8-week-old male, $n = 6$/group) were daily gavaged with vehicle (0.5% methyl cellulose), bafetinib (20 mg/kg), or entospletinib (10 mg/kg) at 7 pm and subjected to HFD feeding for 8 weeks. **b** Body weight of the mice was monitored every week. **c**, **d** Gonadal WAT was subjected to H&E staining, and representative pictures were shown (**c**). Scale bar, 100 μm. Quantification of the surface area of adipocytes in gWAT by ImageJ. The size of each adipocyte was quantified by ImageJ and plotted as mean ± SEM from 287, 488, 305, 324, 362, and 218 cells, respectively **d**. Two-sided Student's *t* test was performed between vehicle and bafetinib (Baf.), or entospletinib (Ent.) treated group, respectively. These experiments were repeated twice. Source data are provided as a Source Data file.

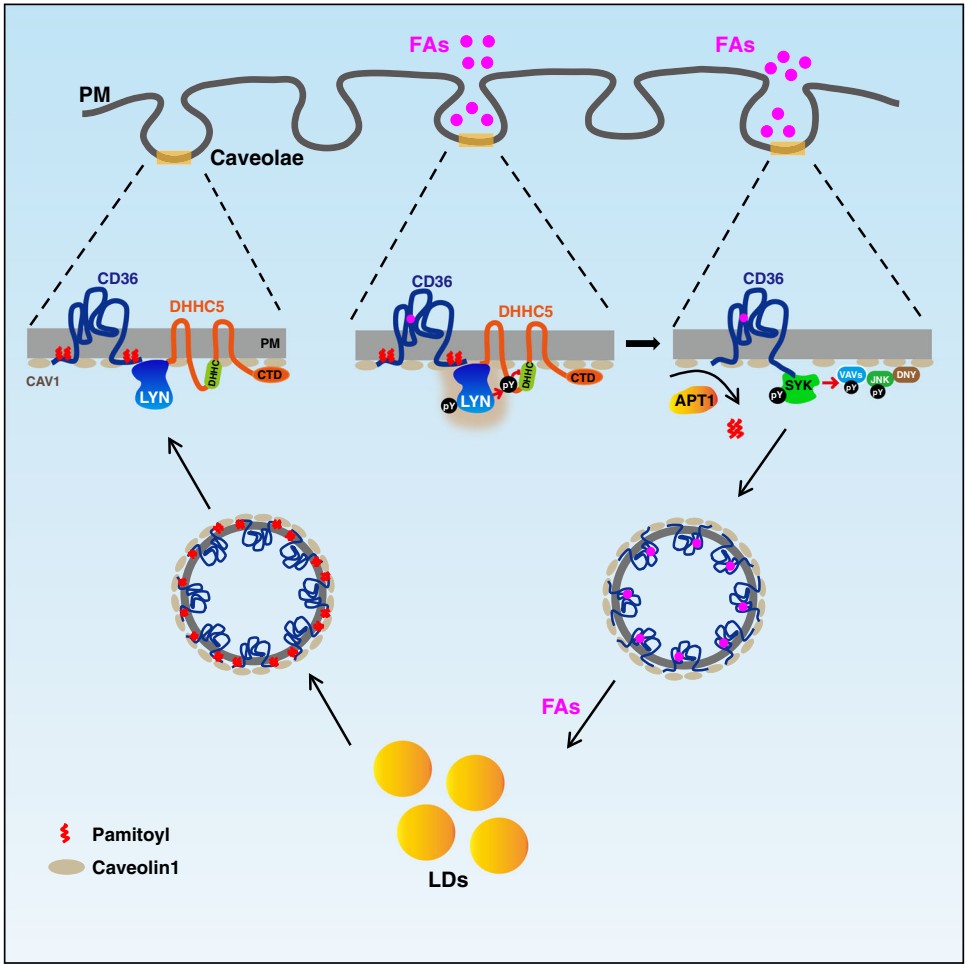

**Fig. 8 A working model of FA uptake by CD36-mediated caveolar endocytosis.** The plasma membrane of adipocytes is enriched with caveolaes. In a caveolae, CD36 is in the palmitoylated form and its FA binding cavity is on the outer layer, whereas CAV1 is on the inner layer. When FAs bind to CD36, a SRC kinase LYN gets activated, and it phosphorylates DHHC5 at Tyr91 and inactivates DHHC5, resulting in the subsequent depalmitoylation of CD36 by the depalmitoylase APT1. The depalmitoylated CD36 then recruits SYK to phosphorylate VAV and JNK, thereby initiating CD36-mediated caveolar endocytosis. The endocytosed vesicles deliver FAs to lipid droplets for storage. After that, CD36 gets re-palmitoylated and recycled to the plasma membrane for another round of delivery.

harvested and immunoprecipitated with anti-Flag M2 beads. Meanwhile, about 3 g gWAT were collected from mice that were fasted and refed with HFD for 4 h. Tissues were then homogenized in Buffer B, and the resultant supernatant was incubated with the M2 beads at 4 °C for 4 h. The beads were eluted with 3 × Flag peptides (0.2 mg/ml), and the eluted proteins were separated on SDS-PAGE, and subjected to silver staining and mass spectrometry analysis.

**Surface biotinylation**. Briefly, cells were incubated with freshly prepared sulfo-NHS-SS-biotin (0.5 mg/ml) in PBS for 30 min on ice. Cells were then washed three times with ice-cold PBS containing 50 mM glycine. Cells were lysed in PBS containing 1% triton X-100, and supernatant was incubated with streptavidin agarose at 4 °C for 2 h. Beads were washed five times with PBS containing 1% triton X-100, eluted with SDS loading buffer at 37 °C for 1 h, and subjected to western blot analysis.

**Immunofluorescence**. After indicated treatments in the figure legends, cells were fixed with 4% paraformaldehyde for 15 min at room temperature, permeabilized with 0.05% NP40 and 0.05% saponin in PBS for 5 min, and blocked with 5% BSA in PBS for 1 h. Cells were then incubated with anti-CD36 (1:100, Abcam, ab23680), anti-CAV1 (1:100, CST, 3238s), anti-Flag antibody (1:100, Sigma-Aldrich, F1804), or anti-HA (1:100, Roche, 11867423001) overnight at 4 °C followed by three times wash with PBS, and then incubated with Alexa Fluor 488 goat anti-rabbit IgG secondary antibody (1:200, Thermo Fisher Scientific) or Alexa Fluor 594 goat anti-mouse IgG secondary antibody (1:200, Thermo Fisher Scientific) for 3 h at 4 °C followed by another three washed with PBS. Cells were then stained with LipidTOX (Thermo Fisher, H34477) and washed three times with PBS. Cover slides were then mounted. Images were taken using a Zeiss LSM-780 confocal microscopy. Imaris 9.2.0 was used for 3D reconstruction.

**PacFA labeling of 3T3-L1 adipocytes**. PacFA labeling was modified from a previous study[23]. 3T3-L1 adipocytes were pretreated with serum-free medium for 4 h, and then treated with BSA or BSA-conjugated PacFA (50 μM) for 20 min. Cells were washed three times with PBS containing 1% BSA and subjected to UV crosslinking on ice for 30 min.

For fluorescent detection of PacFA, cells were fixed with methanol at −20 °C for 20 min and then subjected to click chemistry in 1 mM ascorbic acid, 0.1 mM TBTA, 1 mM CuSO4, and 2 μM Alexa 488 azide (Thermo Fisher, A10266) at 25 °C for 1 h. Cells were then washed with PBS and subjected to immunostaining of CD36 as above.

For isolation of PacFA-labeled proteins, cells were lysed in PBS containing 1% SDS supplemented with protease inhibitors and subjected to click chemistry in 1 mM ascorbic acid, 0.1 mM TBTA, 1 mM $CuSO_4$, and 1 mM biotin azide (Thermo Fisher, B10184) at 25 °C for 1 h. To remove free biotin azide, cell lysate was precipitated with cold acetone (three volumes) at −20 °C for 1 h, and the resultant pellet was washed twice with cold 70% acetone. Pellet was dried, resuspended in PBS containing 0.1% SDS, and incubated with streptavidin beads at 4 °C for 1 h. Beads were washed three times with PBS containing 0.1% SDS and then eluted with Buffer D (50 mM Tris, pH7.4, 150 mM NaCl, 2% SDS, 0.1 M Urea, 5 mM EGTA, 5 mM EDTA, 5 mM biotin, and 10% DMSO). The eluted fraction was then subjected to immunoblotting with indicated antibodies.

**Fatty acid uptake**. For FA uptake in preadipocytes, cells were pretreated in serum-free medium for 4 h, or PP2 (20 μM), ML348 (10 μM), or piceatannol (40 μM) for 1 h, followed by treatment with 100 μM BSA-conjugated oleate for 4 h. Cells were then stained with BODIPY 493/503 (Thermo Fisher, D3922). Images were taken under a Zeiss LSM-780 confocal microscopy using an excitation wavelength of 488 nm with the same laser intensity. None of the images were overexposed.

Fluorescent intensity of BODIPY was quantified from 20 cells of each treatment using ZEN 2010.

For $^3$H-oleate uptake in adipocytes, cells were pretreated with the same inhibitors as above, and then treated with 100 µM BSA-conjugated $^3$H-oleate (specific activity, 2268 dpm/nmol) for 1 h. Cells were washed twice with cold PBS containing 0.2% BSA twice, once with cold PBS, and extracted with hexane: isopropanol (3:2). The extracted fractions were subjected to scintillation counting. After extraction, the cells on the dish were dried out for measurement of protein content by a Pierce BCA protein assay kit.

**Growth of lipid droplets**. Adipocytes were pretreated with serum-free medium for 4 h, and indicated inhibitors for 1 h in the figure legends. Cells were then treated with BSA-conjugated oleate (100 µM) and BODIPY 493/503 (0.1 µg/ml), and subjected to live imaging on a Zeiss LSM-780 confocal microscope in a 3D Z-stack mode for 2 h. In all, 8–12 cells of each treatment were imaged. Images were reconstructed with Imaris 9.2.0, and volume of lipid droplets per cell was calculated.

**Bafetinib and entospletinib treatment and HFD feeding**. WT and $Cd36^{-/-}$ mice (8-week-old male, $n = 6$/group) were daily gavaged with vehicle (0.5% methyl cellulose), bafetinib (20 mg/kg), or entospletinib (10 mg/kg) at 7 p.m. and subjected to HFD (Research Diet, D12492) feeding for 8 weeks. Glucose tolerance test was performed on week 8 as previously described[20]. At the end of the experiment, mice were dissected and H&E staining of gWAT was performed as previously described[20].

**Quantitative real-time PCR**. Total RNA was isolated using Trizol, and quantitative real-time PCR was performed according to Zhao et al.[57]. Briefly, RNA was reversed transcribed using a kit form Toyobo (FSQ-301), and real-time PCR was performed using a SYBR green master mix (Yeasen, 11203ES08). The primers are listed in Supplementary Table 1. The mRNA of 36B4 was used as the invariant control.

**Electron microscopy**. 3T3-L1 adipocytes were cultured and differentiated as above. After trypsinization, cells were fixed in 2.5% glutaraldehyde for 2–3 h at 4 °C. After fixation, three volumes of 0.1 M PBS were added to the fixation buffer, and cells were washed three times with 0.1 M PBS. Samples were then postfixed with 1% OsO4, embedded, and sectioned. Specimens were visualized on a Hitachi HT-7800 transmission election microscope.

**Quantification and statistical analysis**. All the statistical analysis was performed by Student's two-tailed paired $t$ test using EXCEL2010. All the values represent mean ± SEM. All the statistical details of the experiments can be found in the figure legends, including exact number of cells or mice. No data were excluded from any of the experiments.

## Data availability

All data generated or analyzed during this study are included in this published article and its supplementary information files. Source data are provided with this paper.

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

## Acknowledgements

We thank Drs. Peng Li and Bao-Liang Song for critical reading of the paper. This work was supported by the National Key R&D Program of China (2016YFA0502003 to T.-J.Z.), the National Natural Science Foundation of China (91857107, 31671223, and 31871193 to T.-J.Z; 81772818 to H.G), the Fundamental Research Funds for the Central Universities (20720180046 to H.G.), Project "111" sponsored by the State Bureau of Foreign Experts and Ministry of Education of China (BP2018017).

## Author contributions

J.-W.H., J.W., H.G., H.-H.S., Y.-Y.Z., Y.-F.L., X.-Y.L., N.Z., and X.W. performed the experiments. C.X., L.H., X.H., H.-R.W., C.-B.L., and B.L. provided expertise and materials. S.C. and T.-J.Z. designed the experiments, analyzed the data, and wrote the paper.

## Competing interests

The authors declare no competing interests.
