## [Peer Review File · Nature Communications]

Reviewers' comments:

Reviewer #1 (Remarks to the Author):

This study is a nice follow up on a previous report (reference 20) by the same group of investigators documenting importance of CD36 palmitoylation in fatty acid (FA) uptake. In the present manuscript, the authors show that in adipocytes FA internalize CD36 from caveolae together with Cav1 and that dynamic CD36 depalmitoylation is needed for FA-induced CD36 internalization. Using adipocytes and other cells they rely primarily on microscopy to show CD36 internalization and co-localization of the FA with internalized CD36 around lipid droplets. The authors also propose that Lyn kinase, VAV and JNK are involved in CD36 internalization by exogenously added FA. The studies provide interesting mechanistic information on how depalmitoylation influences CD36 internalization by FA. However, the data often are not robust enough, more quantification and important controls are needed. The endocytic process itself is treated superficially and not well characterized and the in vivo studies are difficult to interpret because they lack basic characterization.

Comments:

1- In all figures involving microscopy: More than one cell should be shown in the figure panels and some form of quantification should be included.

With all adipocyte image panels considering the difficulty of showing internalization in these cells as a result of the large lipid droplets, including a membrane marker could be helpful. For example in figure 1e the C16 and C20:4 images show what appears to be relatively little CD36 internalization while the blot in panel f shows that CD36 internalization was identical with all FA tested. In addition, it would be helpful to include a movie in the supplement on the 3D Z-stack reconstruction.

2- In Figure 1 surface biotinylation is used to monitor membrane depletion of CD36 and Cav1. Why is Cav1 biotinylated since it is present on the internal leaflet of the membrane? As a result how quantitative is Cav1 biotinylation. Considering that adipocytes have abundant Cav1 expression it is hard to imagine Cav1 depletion by CD36 internalization. The authors should address this technically. In addition they should provide other membrane proteins as controls in the western blots. Do the adipocytes show fewer caveolae at 1h after FA. Maybe electron microscopy would be helpful.

3- In Figure 2, the PacFA is used to show that the FA is co-localized with CD36 and co-localization is shown to average 60% (panel b). However, the images presented appear to show less than 60% localization, suggesting that more cells need to be shown and more than 10 cells should be counted. -There are two controls that need to be provided: Show that PacFA is similar to native FA in terms of processing by adipocytes, and include data on specificity of the click reaction in targeting FA interacting proteins (and not proteins that are not thought to interact with FA). -It would also be informative here to include CD36^{-/-} adipocytes.

4- In Figure 2 panel d: why is a 1h timepoint used for oleate uptake when CD36 membrane depletion is almost complete at that time. In addition, figures 5 and 6 show that many of the steps involved in CD36 internalization occur within 5min. What is the early time course of FA uptake? Is the labeled FA assay being used only as a marker of lipid droplets?

5- In figure 3 panels c and d, the efficiency of APT1 depletion in adipocytes should be shown. The related data presented in the supplement are not sufficiently convincing. Show more cells in panel b and strengthen the documentation of internalization as mentioned under comment 1.

6- In figures 5 and 6 more immunostained cells should be shown. Figure legends state that the experiments were repeated at least twice. Some quantification of internalization is needed. Again uptake is measured at 1h which seems to be out of sync with the events regulating endocytosis. A better time course of endocytosis should be presented.

-In figure 6 it is presumed that depalmitoylation of CD36 and recruitment of Syk are occurring at the membrane since they are proposed to be important for endocytosis to occur. Can Syk translocation and recruitment be directly documented?

7-Figure 7: The in vivo data are difficult to interpret as showing the effect of blocking FA uptake in adipocytes. No data are presented to confirm that food consumption and absorption (postprandial lipids) are similar between groups. Circulating lipid levels are needed. If these parameters are the same then where is the lipid going? Liver, muscle?

In addition body composition would be nice to confirm that the change in weight is due to impaired adiposity.

8- In the discussion, the authors should provide some explanation of how SYK, VAV and JNK initiate endocytosis.

minor comment: what do the authors mean by "binding of FAs to CD36 activates its downstream kinase LYN, thereby passing the signal from the cell surface to the intracellular depots"?

Reviewer #2 (Remarks to the Author):

This study addresses the mechanism and regulation of fatty acid uptake by CD36. The authors demonstrate that removal of CD36 from the cell surface is stimulated in the presence of fatty acids in a process that is dependent upon caveolin. The authors uncover a cycle of palmitoylation and depalmitoylation mediated respectively, by the palmitoyltransferase DHH5 and the depalmitoylase APT1, that is required for CD36 internalization and fatty acid uptake. The authors identify two tyrosine kinases, Lyn and Syk, that are activated downstream of fatty acid binding to CD36. Lyn phosphorylates DHH5, enabling depalmitoylation of CD36 and subsequent recruitment of Syk, which in turn recruits the cellular machinery required for internalization of CD36. The physiological relevance of this new pathway is demonstrated by showing the effects of an endocytosis block on lipid droplet growth in adipocytes and high-fat-diet induced weight gain in mice.

The study will be of interest in the fields of protein lipidation and lipid metabolism. The authors uncover the mechanism that underlies a regulatory cycle of palmitoylation and depalmitoylation with an important physiological impact.

Overall, the study is technically sound and in most cases, the data support the conclusions drawn by the authors. However, there are some shortcomings that should be addressed. These are outlined as follows.

1. The authors identify APT1 as the depalmitoylase of CD36. This was somewhat surprising in that APT1 is reported to be localized in the mitochondria (Kathayat RS et al. Nat Commun. 2018 9:334. doi: 10.1038/s41467-017-02655-1.) Candidate depalmitoylases other than APT1 were excluded only on the basis of CD36 localization after knockdown of individual depalmitoylases. A more rigorous approach to testing for the involvement of other depalmitoylases should be taken.

2. In some cases, poor image quality precludes the reader from assessing the data. This is true for Figure 3h, Figure 4h. Better images should be presented.

3. The conclusion that DHH5 is inactivated by phosphorylation at Tyr91 would be strengthened by showing that palmitoylation of a second substrate is affected. Flotillin-2 would be a good candidate (J Biol Chem. 2012 287:523-30.)

4. In Figure 5b, how expression of SRC family kinases was assessed should be stated or provided in the Methods section.

5. Cell numbers were not provided for the quantitation of Figure 5j. Images are of poor quality. More information on how BODIPY intensity is quantified should be provided in the methods section.

6. Knockdown of Src-family kinases should be confirmed by showing reduced protein levels – at least for Lyn kinase.

Reviewers' comments:

Reviewer #1 (Remarks to the Author):

This study is a nice follow up on a previous report (reference 20) by the same group of investigators documenting importance of CD36 palmitoylation in fatty acid (FA) uptake. In the present manuscript, the authors show that in adipocytes FA internalize CD36 from caveolae together with Cav1 and that dynamic CD36 depalmitoylation is needed for FA-induced CD36 internalization. Using adipocytes and other cells they rely primarily on microscopy to show CD36 internalization and co-localization of the FA with internalized CD36 around lipid droplets. The authors also propose that Lyn kinase, VAV and JNK are involved in CD36 internalization by exogenously added FA. The studies provide interesting mechanistic information on how depalmitoylation influences CD36 internalization by FA. However, the data often are not robust enough, more quantification and important controls are needed. The endocytic process itself is treated superficially and not well characterized and the in vivo studies are difficult to interpret because they lack basic characterization.

Comments:

1- In all figures involving microscopy: More than one cell should be shown in the figure panels and some form of quantification should be included. With all adipocyte image panels considering the difficulty of showing internalization in these cells as a result of the large lipid droplets, including a membrane marker could be helpful. For example in figure 1e the C16 and C20:4 images show what appears to be relatively little CD36 internalization while the blot in panel f shows that CD36 internalization was identical with all FA tested. In addition, it would be helpful to include a movie in the supplement on the 3D Z-stack reconstruction.

Response: Thank you for your suggestions. Our original thought to show one cell per image was to make the endocytosis more visible. According to the suggestion, we have

re-performed all the immunostaining assay and included 2 cells or more in each image. To better illustrate the results, we have also enlarged one cell in each image in the revised manuscript.

To show the extent of internalization quantitatively, we quantified the surface content of CD36 in the surface biotinylation assay. Each quantification was performed from 3 individual experiments, and the quantified data are now included in the revised manuscript.

As the reviewer suggested, we screened some of the reported plasma membrane markers and found that ATP1A1, a subunit of Na⁺/K⁺-ATPase, showed a good immunostaining signal and it stayed on the plasma membrane under both basal and FA-treated conditions. We therefore used ATP1A1 as a plasma membrane marker and have clearly demonstrated that CD36 is indeed internalized after fatty acid treatment. In Fig. 1a and Fig. 1c where we performed immunostaining using anti-CD36 and anti-CAV1 antibodies, as the primary antibodies of anti-CAV1 and anti-ATP1A1 are both rabbit antibodies, we could not immunostain ATP1A1 at the same time. For the rest of the panels, we immunostained both CD36 and ATP1A1, and we could clearly see the fatty acid-induced endocytosis.

As suggested, we have now included a movie in the supplement on the 3D Z-stack reconstruction.

2- In Figure 1 surface biotinylation is used to monitor membrane depletion of CD36 and Cav1. Why is Cav1 biotinylated since it is present on the internal leaflet of the membrane? As a result how quantitative is Cav1 biotinylation. Considering that adipocytes have abundant Cav1 expression it is hard to imagine Cav1 depletion by CD36 internalization. The authors should address this technically. In addition they should provide other membrane proteins as controls in the western blots. Do the adipocytes show fewer caveolae at 1h after FA. Maybe electron microscopy would be helpful.

Response: Thank you for the question. As the reviewer pointed out, CAV1 is in the

inner leaflet of caveolae, and CD36 is mainly on the outer membrane. It is known that the caveolae are detergent resistant; therefore, when we solubilize the cells in 1% Triton in surface biotinylation as described in the *Method* session, the caveolae structures will not get disrupted. In that way, when we pulled down biotinylated CD36, CAV1 will also get pulled down. Although we have stated this clearly in the Methods, to avoid any unnecessary confusion, we decided to take CAV1 out from the surface biotinylation results.

In terms of including a plasma marker as a control for Western blots, we examined the surface content of ATP1A1 by the surface biotinylation assay. Consistent with the immunostaining results, surface content of ATP1A1 did not change before and after fatty acid treatment. We have now included ATP1A1 as a plasma marker in all the surface biotinylation experiments.

As suggested, we performed electron microscopy. Consistent with our immunostaining results, the caveolae were mainly localized on the plasma membrane in BSA-treated cells, but the majority of them were internalized 1 hr after oleate treatment. The figure has now been included in Supplementary Fig. 1b.

3- *In Figure 2, the PacFA is used to show that the FA is co-localized with CD36 and co-localization is shown to average 60% (panel b). However, the images presented appear to show less than 60% localization, suggesting that more cells need to be shown and more than 10 cells should be counted.*

-There are two controls that need to be provided: Show that PacFA is similar to native FA in terms of processing by adipocytes, and include data on specificity of the click reaction in targeting FA interacting proteins (and not proteins that are not thought to interact with FA).

-It would also be informative here to include CD36^{-/-} adipocytes.

Response: Thank you for your suggestions. We have re-performed the experiment and quantified colocalization of PacFA and CD36 in 24 cells. As shown in the new Fig. 2b, the co-localization is still about 56%. We have also replaced Fig. 2a with two cells.

In the original paper about PacFA, Haberkant et al. has clearly demonstrated that PacFA acts similar to native FA and can be incorporated into diacylglycerol, triacylglycerol, cholesterol ester and phospholipids in HeLa and CHO cells (Angew Chem Int Ed Engl, 2013, 52, 4033-4038). We did try to perform similar experiments by ordering the reagents including the 3-azido-7-hydroxycoumarin that is a key reagent in the assay. Due to the breakout of COVID-19, although we submitted the order in early March, the reagent has not come in yet. Considering that the fatty acid esterification and triglyceride and phospholipid synthesis enzymes are well evolutionarily conserved from yeast to human, it is reasonable to believe that PacFA will act similar to native FA in adipocytes too. We therefore ask not to perform the experiment.

To add control on the specificity of the click reaction in targeting FA interacting proteins, we tested whether PacFA would bind fatty acid binding protein 4 (FABP4), a known FA interacting protein. As shown in Fig 2c, we found that FABP4 could also be pulled down by PacFA, which confirms that the click chemistry reaction is specific.

In terms of performing PacFA treatment in *Cd36*^{-/-} adipocytes, we need to mention that the major purpose of using PacFA was to demonstrate the co-migration of FAs with CD36. This experiment was not designed for quantification purpose, as UV crosslinking and click chemistry could possibly generate too much variance in the quantitative assay. To quantify FA uptake, we used ³H-oleate, which is more quantitative. As shown in Fig. 2d, *Cd36*^{-/-} adipocytes show about 40% decrease in FA uptake activity. If we perform the experiment, *Cd36*^{-/-} adipocytes were expected to show PacFA uptake signal, but not much would be learned from this experiment. As this would not be a quantitative assay and we have performed FA uptake using other methods, we therefore ask not to perform the experiment.

4- In Figure 2 panel d: why is a 1h timepoint used for oleate uptake when CD36 membrane depletion is almost complete at that time. In addition, figures 5 and 6 show that many of the steps involved in CD36 internalization occur within 5min. What is the early time course of FA uptake? Is the labeled FA assay being used only as a marker of lipid droplets?

Response: Thank you for the question. In Fig. 1a,b and supplementary Fig. 1c, we have clearly demonstrated that the internalization of CD36 did not start until 30 min after oleate treatment. Therefore, in order to measure CD36-mediated endocytosis of fatty acids, it would be better to analyze CD36-mediated endocytosis of FAs at least 30 min after ³H-oleate treatment. We have actually performed a time course curve of ³H-oleate uptake and found fatty acid uptake was linear in 1 hr (See the figure below). That is why we chose the 1-hr timepoint for the ³H-oleate uptake experiment.

As the reviewer pointed out, the LYN-SYK-JNK signaling started within 5 min of oleate treatment (Fig. 5 and Fig. 6), the endocytosis did not start until 30 min. The delay in endocytosis is likely because it takes time to convert the signaling pathway into endocytosis, which requires cytoskeleton reorganization and pinch off of the caveolae from the plasma membrane.

As mentioned in Comment#3, PacFA was mainly used to indicate the co-migration of CD36 and FAs, not for quantification purpose.

Figure 1. Time course of 3H-oleate uptake in 3T3-L1 adipocytes.

5- In figure 3 panels c and d, the efficiency of APT1 depletion in adipocytes should be shown. The related data presented in the supplement are not sufficiently convincing. Show more cells in panel b and strengthen the documentation of internalization as mentioned under comment 1.

Response: Thank you for your suggestions. We have now included the knockdown

efficiency of APT1 in Fig. 3c and 3d.

We have also performed more experiments to strength the supplementary data. First, we replace the immunostaining images in supplementary Fig. 3a and 3h to two or more cells per image. Second, we tested the effect of all of the 5 reported depalmitoylases on CD36 palmitoylation, and found that only APT1 could dramatically decrease the palmitoylation of CD36. Combined the other experiments in Fig. 3, our data are convincing that APT1 is the depalmitoylase of CD36.

For Fig. 3b and 3e, we have performed new experiment using ATP1A1 to indicate the plasma membrane, and included two cells in these two panels. We have also quantified the surface content of CD36, and the results are now included in Supplementary Fig. 3i and 3j.

6- In figures 5 and 6 more immunostained cells should be shown. Figure legends state that the experiments were repeated at least twice. Some quantification of internalization is needed. Again uptake is measured at 1h which seems to be out of sync with the events regulating endocytosis. A better time course of endocytosis should be presented.

-In figure 6 it is presumed that depalmitoylation of CD36 and recruitment of Syk are occurring at the membrane since they are proposed to be important for endocytosis to occur. Can Syk translocation and recruitment be directly documented?

Response: Thank you for your suggestions. We have performed new experiment and included two cells in each panel in Fig. 5 and Fig. 6. We have also quantified the surface content of CD36 by surface biotinylation assay and included them in Supplementary Fig. S5 and S6. In terms of choosing the 1-hr timepoint, please refer to our response to comment #4.

As the reviewer suggested, we tried to document the translocation of SYK to the membrane. As the SYK antibody we had was not good for immunostaining, we isolated membrane fraction before and 5 min after oleate treatment and detected SYK by Western blot. We found that SYK was not detected at the membrane fractions before oleate treatment, but it was readily detected 5 min after treatment. We have now included the data in Supplementary Fig. 6d.

7-Figure 7: The in vivo data are difficult to interpret as showing the effect of blocking FA uptake in adipocytes. No data are presented to confirm that food consumption and absorption (postprandial lipids) are similar between groups. Circulating lipid levels are needed. If these parameters are the same then where is the lipid going? Liver, muscle?

In addition body composition would be nice to confirm that the change in weight is due to impaired adiposity.

Response: Thank you for your suggestions. We had the data of food intake and plasma lipid levels. Neither bafetinib or entospletinib treatment had any effect on the food intake, plasma free fatty acid and plasma triglyceride in WT or *Cd36*^{-/-} mice. We have now included the data in Supplementary Fig. 9b, d,e.

To figure out where the lipid is going, we measured liver content of triglyceride and found that both compounds slightly but significantly increased liver triglyceride levels in WT mice, suggesting that some of the lipids were ectopically stored in liver. These data are now included in Supplementary Fig. 9f.

In terms of adiposity, we actually had the data included in the original manuscript (Supplementary Fig. 8b, now Supplementary Fig. 9c). The fat mass was significantly lower in WT mice treated with either bafetinib or entospletinib. Therefore, the decrease in body weight was due to impaired adiposity.

8- In the discussion, the authors should provide some explanation of how SYK, VAV and JNK initiate endocytosis.

minor comment: what do the authors mean by "binding of FAs to CD36 activates its downstream kinase LYN, thereby passing the signal from the cell surface to the intracellular depots"?

Response: Thank you for your suggestions. We have added the following sentences to explain the roles of VAV and JNK. "VAV function as an adaptor of dynamin^{32, 33}, which facilitates the pinching off of caveolae from the plasma membrane²¹. The activated JNK plays an important role in regulating cytoskeleton re-organization and vesicle

transport⁵², two key events caveolar endocytosis.”.

Thank you, and we apologize for the confusion about the way we wrote up the sentence. We have now changed the sentence to “binding of FAs to CD36 activates its downstream kinase LYN, thereby converting the extracellular stimulus of FAs into intracellular signaling pathway”.

Reviewer #2 (Remarks to the Author):

This study addresses the mechanism and regulation of fatty acid uptake by CD36. The authors demonstrate that removal of CD36 from the cell surface is stimulated in the presence of fatty acids in a process that is dependent upon caveolin. The authors uncover a cycle of palmitoylation and depalmitoylation mediated respectively, by the palmitoyltransferase DHHC5 and the depalmitoylase APT1, that is required for CD36 internalization and fatty acid uptake. The authors identify two tyrosine kinases, Lyn and Syk, that are activated downstream of fatty acid binding to CD36. Lyn phosphorylates DHHC5, enabling depalmitoylation of CD36 and subsequent recruitment of Syk, which in turn recruits the cellular machinery required for internalization of CD36. The physiological relevance of this new pathway is demonstrated by showing the effects of an endocytosis block on lipid droplet growth in adipocytes and high-fat-diet induced weight gain in mice.

The study will be of interest in the fields of protein lipidation and lipid metabolism. The authors uncover the mechanism that underlies a regulatory cycle of palmitoylation and depalmitoylation with an important physiological impact.

Overall, the study is technically sound and in most cases, the data support the conclusions drawn by the authors. However, there are some shortcomings that should be addressed. These are outlined as follows.

1. *The authors identify APT1 as the depalmitoylase of CD36. This was somewhat surprising in that APT1 is reported to be localized in the mitochondria (Kathayat RS et al. Nat Commun. 2018 9:334. doi: 10.1038/s41467-017-02655-1.) Candidate depalmitoylases other than APT1 were excluded only on the basis of CD36 localization after knockdown of individual depalmitoylases. A more rigorous approach to testing for the involvement of other depalmitoylases should be taken.*

Response: Thank you for your suggestions. As the reviewer pointed out, Kathayat Rs et al. reported that APT1 is primarily localized in the mitochondria (Nature Commun, 2018). In another report by Stypulkowski et al., APT1 is also localized on plasma membrane (Science Signaling, 2018). To further clarify whether other depalmitoylases (APT2, ABHD17A, ABHD17B and ABHD17C) could depalmitoylate CD36, we co-transfected each of them with CD36 and examined their effect on CD36 palmitoylation. As shown in Supplementary Fig. 3c-g, only APT1, but not the others, dramatically reduced the palmitoylation of CD36. These results further strengthen our point that APT1 is the depalmitoylases of CD36.

2. *In some cases, poor image quality precludes the reader from assessing the data. This is true for Figure 3h, Figure 4h. Better images should be presented.*

Response: Thank you, and we apologize for that. For Fig. 3h, the images were indeed too small in the previous submission. We have now enlarged the images in the revised manuscript, and we can clearly see the difference in BODIPY staining in different images. For Fig. 4h, we performed new experiments and replaced the figure. We can clearly see that WT DHHC5, but not Y91E or Y91F mutant, promotes FA uptake in DHHC5 knockdown cells.

3. *The conclusion that DHHC5 is inactivated by phosphorylation at Tyr91 would be strengthened by showing that palmitoylation of a second substrate is affected. Flotillin-2 would be a good candidate (J Biol Chem. 2012 287:523-30.)*

Response: Thank you for your suggestion. As the reviewer suggested, we examined the activity of Y91E and Y91F mutants of DHHC5 in palmitoylating Flotillin-2. As shown in Supplementary Fig. 4, the Y91E mutant showed much decreased activity in palmitoylating Flotillin-2, consistent with our observation using CD36 as the substrate. These results further confirm that phosphorylation at Y91 inactivates DHHC5.

4. In Figure 5b, how expression of SRC family kinases was assessed should be stated or provided in the Methods section.

Response: Thank you for your careful reading. The expression of the SRC family kinases were analyzed by a RNA-Seq study in white adipose tissue. We have now stated it in the manuscript as “Images were taken under a Zeiss LSM-780 confocal microscopy using an excitation wavelength of 488 nm with the same laser intensity. None of the images were overexposed.

5. Cell numbers were not provided for the quantitation of Figure 5j. Images are of poor quality. More information on how BODIPY intensity is quantified should be provided in the methods section.

Response: Thank you for your suggestions. We quantified 20 cells per treatment in Fig. 5j, and we have included this in the figure legends. Again, we apologize for making Figure 5 h too small in the previous submission. We have now enlarged the images and we can clearly see the difference in the intensities of BODIPY in different images. We have also added more details of how to quantify BODIPY intensity in the methods section as “Images were taken under a Zeiss LSM-780 confocal microscopy using an excitation wavelength of 488 nm with the same laser intensity. None of the images were overexposed.”.

6. Knockdown of Src-family kinases should be confirmed by showing reduced protein levels – at least for Lyn kinase.

Response: Thank you for your suggestion. We have confirmed the knockdown

efficiency of LYN by Western blot and updated them in Fig. 5f and 5g.

Reviewer #1 (Remarks to the Author):

The authors did a great job in revising their manuscript. They have addressed all comments appropriately and provided additional data to strengthen their findings especially as related to the microscopy.

Minor Comments:

- 1- With respect to Cav-1 the authors chose to remove the biotinylation data, which was not necessary. Perhaps they should keep these data while explaining them, so others know that Cav-1 can get biotinylated under certain conditions.
- 2- The electron microscopy is a great addition, although the EM could be improved.
- 3- The CD36 mutant K164 did not internalize but it is not clear from the figure presented. Did oleate lead to formation of lipid droplets in K164 cells?
- 4- On page 6 the authors state "we wondered whether CD36 and CAV1 would depend on each other for the internalization in a way that CD36 acts like an antenna to receive the signal from FAs". It would be better to substitute "antenna" with "receptor" since CD36 internalizes with the FA.

Reviewer #2 (Remarks to the Author):

The study will be of interest in the fields of protein lipidation and lipid metabolism. The authors uncover the mechanism that underlies a regulatory cycle of palmitoylation and depalmitoylation with an important physiological impact. The revised manuscript satisfactorily addresses the issues raised in my review.

REVIEWERS' COMMENTS:

Reviewer #1 (Remarks to the Author):

The authors did a great job in revising their manuscript. They have addressed all comments appropriately and provided additional data to strengthen their findings especially as related to the microscopy.

Minor Comments:

1- With respect to Cav-1 the authors chose to remove the biotinylation data, which was not necessary. Perhaps they should keep these data while explaining them, so others know that Cav-1 can get biotinylated under certain conditions.

Response: Thank you. We have added the data back to Figure 1, and we added a short sentence to explain it.

2- The electron microscopy is a great addition, although the EM could be improved.

Response: Thank you. We have tried our best for the current study. We will work on our EM skills in the future.

3- The CD36 mutant K164 did not internalize but it is not clear from the figure presented. Did oleate lead to formation of lipid droplets in K164 cells?

Response: Thank you. We apologized for not labeling what the fluorescent colors stand for in Supplementary Fig. S1d, and we have now added the label to the figure. We can clearly see that in oleate-treated K164R mutant cells CD36 and ATP1A1 colocalized on the plasma membrane and did not internalize. As mentioned in the earlier revision, CD36 accounts for about 50%, not all, of the FA uptake in adipocytes, and we would therefore expect that oleate can still lead to formation of lipid droplets in K164R cells.

4- On page 6 the authors state "we wondered whether CD36 and CAV1 would depend on each other for the internalization in a way that CD36 acts like an antenna to receive the signal from FAs". It would be better to substitute "antenna" with "receptor" since CD36 internalizes with the FA.

Response: Thank you. We substituted "antenna" with "receptor".

Reviewer #2 (Remarks to the Author):

The study will be of interest in the fields of protein lipidation and lipid metabolism. The authors uncover the mechanism that underlies a regulatory cycle of palmitoylation and depalmitoylation with an important physiological impact. The revised manuscript satisfactorily addresses the issues raised in my review.

Response: Thank you very much.